# A GPU-parallelization of the neXtSIM-DG dynamical core (v0.3.1)

Robert Jendersie[1,2], Christian Lessig[1,3], and Thomas Richter[2]

[1]Institute of Simulation and Graphics, Otto-von-Guericke University, Magdeburg, Germany
[2]Institute of Analysis und Numerics, Otto-von-Guericke University, Magdeburg, Germany
[3]European Centre for Medium-Range Weather Forecasts, Bonn, Germany

**Correspondence:** Robert Jendersie (robert.jendersie@ovgu.de)

**Abstract.** The cryosphere plays a crucial role in Earth's climate system, making accurate sea ice simulation essential for improving climate projections. To achieve higher resolution simulations, graphics processing units (GPUs) have become increasingly appealing due to their higher floating point peak performance compared to CPUs. However, harnessing the full theoretical performance of GPUs often requires significant effort in redesigning algorithms and careful implementation. Recently, several frameworks have emerged, aiming to simplify general-purpose GPU programming. In this study, we evaluate multiple such frameworks, including CUDA, SYCL, Kokkos, and PyTorch, for the parallelization of neXtSIM-DG, a finite-element-based dynamical core for sea ice. Based on our assessment of usability and performance, CUDA demonstrates the best performance, while Kokkos is a suitable option for its robust heterogeneous computing capabilities. Our complete implementation of the momentum equation using Kokkos achieves a sixfold speedup on the GPU compared to our OpenMP-based CPU code, while maintaining competitiveness when run on the CPU. Additionally, we explore the use of lower precision floating-point types on the GPU, showing that switching to single precision can further accelerate sea ice codes.

## 1 Introduction

Simulations are essential for understanding the effects of climate change and enabling stakeholders to mitigate its impact on societies and individuals (Jakob et al., 2023). The cryosphere is a key component of Earth's climate system and it has a particular impact on long-term processes. neXtSIM-DG is a novel sea ice code that is designed as part of The Scale-Aware Sea Ice Project (SASIP)[1] to improve both the representation of physical processes and the efficiency and accuracy of numerical implementation.

A crucial factor for the fidelity and reliability of climate simulations is horizontal resolution, with kilometer-scale simulations, which e.g. explicitly resolve convection but also many other processes in the Earth system, being the target for the next generation of models (Stevens et al., 2019; Bauer et al., 2021b). These computations require exascale HPC systems with substantial GPU-based accelerators (Schär et al., 2020; Bauer et al., 2021a). As a result, significant efforts have been dedicated to porting components of existing climate models to GPUs (Ikuyajolu et al., 2023; Sauer and Muñoz-Esparza, 2020; Cao et al., 2023; Sun et al., 2023).

---

[1]https://sasip-climate.github.io

In line with these developments, SASIP aims to substantially increase the numerical accuracy of the sea ice component
of coupled climate models by using modern higher order discretizations of the governing equations. Kilometer-scale sea ice
simulations are also of substantial theoretical interest, as initial investigations show a marked change in the behavior of the
commonly used viscous plastic sea ice model when approaching km-scale resolutions (Bouchat et al., 2022; Hutter et al., 2022).
Computationally much more efficient models are needed for detailed investigations of this behavior, its relationship with known
statistical and physical principles (Marsan et al., 2004), and the role of possible alternative models (Dansereau et al., 2016;
Ólason et al., 2022). Kilometer-scale or even higher resolved sea ice forecasts are also in high demand among sea ice forecast
users (e.g. Kauker et al. (2021)), even though it is unclear how current systems should be used at such resolutions (Hunke et al.,
2020). The role of small-scale sea ice features, such as ridges and leads, in atmosphere–ocean–ice interactions in weather and
climate models can also only be speculated on with current 10 km scale resolution models (Esau, 2007; Marcq and Weiss, 2012;
Ólason et al., 2021). Highly efficient kilometer-scale sea ice models, either stand-alone or in fully coupled climate models, are
needed to address these questions.

So far, the GPU parallelization of sea ice models has received limited attention. The usage of GPUs for the finite-difference
based dynamical core of CICE, the Los Alamos sea ice model, has been investigated by Rasmussen et al. (2024).

The neXtSIM-DG dynamical core is a flexible framework to model sea ice using a discretization with a higher order discon-
tinuous/continuous Galerkin method (Richter et al., 2023a). The algorithmic structure is such that different rheologies can be
used. At the core of the implementation is a module that encapsulates the non-linear stress update locally (on each element).
Different rheologies, e.g. viscous plastic (Hibler, 1979) or brittle mechanics (Ólason et al., 2022), can be easily realized with
it. The outer code is a pseudo-time stepping iteration that corresponds to the mEVP iteration (Bouillon et al., 2013), which is
frequently used to approximate the Hibler model and which is essentially identical in all rheologies we consider. To focus on
the GPU implementation, we only discuss the viscous-plastic model with the mEVP solver in this manuscript without loss of
generality. At the heart of the numerical implementation is a large number of identical operations to be carried out on each
mesh element. GPUs, which are based on a data parallel processing model, are well suited for these kinds of computations.

Besides performance, other factors are also important for the implementation of the neXtSIM-DG dynamical core: the code
must be adaptable to future hardware, a long-term support of the software must be guaranteed and it should be easy to use.
Such considerations are vital for the model's adoption and sustained use. With these aspects in mind, we examine the current
landscape of general-purpose GPU programming frameworks and compare prototype implementations for the parallelization
of finite-element/finite-volume codes like neXtSIM-DG.

The de facto standard for general-purpose GPU programming is CUDA (NVIDIA, 2023a), though it only supports NVIDIA
hardware. Additionally, it often necessitates a dedicated GPU implementation, requiring significant effort for development and
performance tuning. As a result, various alternatives have emerged, offering greater flexibility and usability. A focus on ease
of use and minimal effort for porting to GPUs is found in frameworks like OpenMP and OpenACC. These, however, offer less
control and fewer performance optimization options compared to CUDA.

SYCL (The Khronos Group, 2023a) and Kokkos (Trott et al., 2022) are frameworks designed for heterogeneous computing,
enabling one to target of different compute hardware. A major challenge with these frameworks is achieving near-optimal per-

formance across diverse hardware platforms, particularly to efficiently leverage GPU-specific features such as shared memory or tensor cores.

Another recent alternative is the use of libraries such as jax (Bradbury et al., 2018) and PyTorch (Paszke et al., 2019), which were primarily developed for machine learning. Their backends are built upon high-performance linear algebra libraries that support various hardware architectures and include compilers that efficiently map computations onto these platforms. Examples of such backends include XLA (xla, 2023), Triton (Tillet et al., 2019), and TensorRT (NVIDIA, 2023b).

With machine learning being a driving force in the development of new accelerator hardware, significant resources are invested in lower precision floating point support. Since these types offer orders of magnitude higher throughput than double precision, which is commonly used in scientific computing, the use of lower precision datatypes is a promising direction for further speedups. For weather simulations, experiments with a reduced-precision emulator indicates that the precision can be reduced for most variables, in some cases down to half precision, without degrading results (Hatfield et al., 2019; Tintó Prims et al., 2019). For example, the Integrated Forecasting System (IFS) of the European Centre for Medium-Range Weather Forecasts has successfully switched to single precision without requiring major code changes (Lang et al., 2021). Climate simulations with a reduced-precision emulator suggest that single precision can be equally accurate even for long term runs (Paxton et al., 2022; Kimpson et al., 2023; Banderier et al., 2024).

To compare the GPU programming frameworks and to investigate the potential speedup of an optimized GPU implementation, we use the different frameworks to port an important part of the neXtSIM-DG dynamical core to the GPU. Our results show that SYCL is still immature, suffering from an unreliable toolchain. Dedicated CUDA remains the best option for speed, while Kokkos provides comparable performance but greater flexibility. PyTorch is currently not a viable alternative to hand-written C++ code, but the new compiler TorchInductor shows promise. We also find that single precision is a strong option to further accelerate our sea-ice simulation. Based on our evaluation, we use Kokkos to implement the complete momentum equation on GPU, for which we achieve a speedup by a factor of 6 compared to a multi-core CPU implementation.

The structure of the paper is as follows. We start in Sect. 2 with an overview of the neXtSIM-DG dynamical core. The different GPU implementations of it are detailed in Sect. 3. In the subsequent Sect. 4, their performance is compared and the impact of mixed precision as well as higher order discretizations is analyzed. In Sect. 5, the Kokkos implementation is extended and practical results are presented. Finally, directions for future work and a summary is provided in Sect. 6.

## 2   Model description

For simplicity of the presentation, we restrict ourselves to the viscous–plastic sea ice model introduced by Hibler (1979). For a discussion of different material laws we refer to the literature (Feltham, 2008). Full details on the model and its discretization in neXtSIM-DG are given by Richter et al. (2023a).

The model consists of advection equations for the ice height $H$ and ice concentration $A$

$$\partial_t H + \mathrm{div}\,(\mathbf{v}H) = S_H, \qquad \partial_t A + \mathrm{div}\,(\mathbf{v}A) = S_A, \tag{1}$$

where the right-hand side terms $S_H$ and $S_A$ are describing thermodynamics. In addition, the 2d velocity field $\mathbf{v}$ is governed by the momentum equation

$$\rho_{\text{ice}} H \partial_t \mathbf{v} = \operatorname{div} \boldsymbol{\sigma}(\mathbf{v}, A, H) + F, \tag{2}$$

where $\rho_{\text{ice}}$ is the ice density and $\sigma$ the stress tensor. In $F$ we collect all external forcings that come from wind and ocean currents. Implicit solvers for the Hibler model suffer from the strong nonlinearities, are costly and require advanced numerical solution strategies (Mehlmann and Richter, 2017). Hence, often explicit iterations in the sense of a pseudo-time stepping is used, in our case the mEVP solver (Bouillon et al., 2013). This introduces a subcycling index $p$ and the implicit rheology is approximated by iterating the following system consisting of momentum equation and stress update

$$\boldsymbol{\sigma}^{(p)} = \frac{\alpha}{1+\alpha} \boldsymbol{\sigma}^{(p-1)} + \frac{1}{1+\alpha} \boldsymbol{\sigma}_{vp}(\mathbf{v}^{(p-1)}, A, H),$$

$$\boldsymbol{\sigma}_{vp}(\mathbf{v}, A, H) = \eta(\nabla \mathbf{v} + \nabla \mathbf{v}^T) + \zeta \operatorname{div}(\mathbf{v}) I - \frac{P}{2} I \tag{3}$$

$$(1+\beta)\rho_{\text{ice}} H \mathbf{v}^{(p)} = \rho_{\text{ice}} H \left( \mathbf{v}_{n-1} + \beta \mathbf{v}^{(p-1)} \right) + \Delta t \cdot \operatorname{div}(\boldsymbol{\sigma}^{(p)}) + \Delta t F.$$

Here, $\mathbf{v}_{n-1}$ is the velocity from the previous macro time step. The viscosities $\eta, \zeta$ depend on velocity $\mathbf{v}$, ice height $H$ and ice concentration $A$, while the ice strength $P$ depends only on $H$ and $A$. The parameters $\alpha, \beta > 0$ control the stability and the speed of convergence. For an infinite number of subcyling steps $p \to \infty$, the mEVP iteration converges to the implicit VP limit. In practice, usually 100 to 200 subcyling steps are performed, see Kimmritz et al. (2016) for a discussion.

This approach is also the basis of the neXtSIM-DG implementation. While the advection problems Eq. (1) are solved using a large time step, the momentum equation Eq. (2) and the mEVP iteration Eq. (3) are subcycled with a smaller step size. Often more than 100 substeps are required in each advection step and the main effort lies in the repeated evaluation of the nonlinear material law, see $\sigma(\mathbf{v}, A, H)$ in Eq. (3). To use rheologies other than the viscous plastic model, essentially only the local calculation of the stress $\boldsymbol{\sigma}_{vp}^{(p)}$ needs to be modified.

## 2.1 Discretization

We briefly sketch the discretization of the model in the neXtSIM-DG dynamical core. Equations (1) to (3) are discretized on quadrilateral meshes in spherical coordinates. This mesh is topologically fully structured consisting of $N_x \times N_y$ quadrilateral elements. Each element is mapped from a uniform reference element onto the computational element using a bi-linear transformation to allow better alignment with coastlines and a more equal mesh spacing in the ocean and ice covered regions. Hence, the computational mesh consists of general quadrilaterals in lat/lon coordinates. The curvature is accounted for exactly by weighting all integrals (of the dG formulation) with the functional determinant of the spherical coordinate system. For the advection equations higher order discontinuous Galerkin upwind methods and high order explicit Runge-Kutta schemes are used. The velocity $\mathbf{v}$ is discretized using a continuous Galerkin approach. As the velocity stress coupling has the form of a mixed formulation, a discontinuous Galerkin space is used to represent the stresses. This space must include the gradient of the velocity space for stability.

|  | serial [s] | OpenMP (20 threads) [s] | speedup |
|---|---|---|---|
| advection | 188.32 | 34.34 | 5.48 |
| boundary | 38.30 | 11.25 | 3.40 |
| strain | 918.84 | 185.28 | 4.96 |
| stress | **1741.43** | **206.36** | 8.44 |
| divergence | 1023.07 | 170.81 | 5.99 |
| velocity | 728.80 | 85.73 | **8.50** |
| other | 14.42 | 5.98 | 2.41 |
| total | 4653.2 | 699.75 | 6.65 |

**Table 1.** Runtime of 120 time-steps of the simulation with $2.6 \times 10^5$ elements on an Intel i9-10900X (10 cores @ 3.7 GHz). Except for the advection, all major computations are part of the mEVP iteration which performs 100 sub-steps in each time-step. Remaining operations, e.g. i/o and external forcing, are summarized by "other". For OpenMP, simultaneous multithreading is beneficial, as running with 20 threads is 24% faster than 10.

## 3 Implementation

Starting point for the GPU implementation of neXtSIM-DG dynamical core is the C++ CPU implementation that is described in (Richter et al., 2023a). This OpenMP parallelized CPU code also serves as the baseline for the performance evaluation. The CPU implementation leverages the linear algebra library Eigen (Guennebaud et al., 2010), which is highly optimized and, e.g., exploits CPU vector units. Due to the explicit character of the discretization, and the parametric finite element setup, most computations are matrix-vector or matrix-matrix products with small matrices and vectors, e.g. vectors of size 4. As several vectors and matrices are already available at compile time, e.g. all quantities that refer to the reference element, neXtSIM-DG greatly benefits from Eigen's template-based design. These fixed size matrices do not require dynamic memory allocation and operations involving such matrices can be fully loop-unrolled. Also, the use of expression templates in Eigen eliminates unnecessary temporary variables in expressions involving multiple operations.

Table 1 indicates computational times for the different parts of the dynamical core in a typical sea ice dynamics simulation. As a numerical test case for all computations, we used the Mehlmann benchmark problem Mehlmann et al. (2021), which has been tested in various sea ice frameworks since its introduction. It models the impact of a cyclone on a dense ice cover and the formation of linear kinematic features. The square domain is of size $512 \times 512$ km and the simulation typically runs for four days. The mEVP iteration (middle lines of the table from "strain" to "velocity") takes most of the time and the stress update is the single most expensive part. These computations are local on each mesh element and hence scale well with more cores. They are further well suited as computational unit for the evaluation of the different GPU programming frameworks. A pseudocode overview of the stress update computations is shown in Listing 1. The original C++ code is documented in Appendix B. Unless

```
void StressUpdateHighOrder(Matrix<N,n_S>& S^{11}, Matrix<N,n_S>& S^{12}, Matrix<N,n_S>& S^{22},
const Matrix<N,n_S>& E^{11}, const Matrix<N,n_S>& E^{12}, const Matrix<N,n_S>& E^{22},
const Matrix<N,n_A>& H, const Matrix<N,n_A>& A, double α) {
for (i=0; i<N; ++i) { // in parallel
Vector<n_G> h  = max{0, H_{i,*}PSI⟨n_A⟩}
Vector<n_G> a  = min{1, max{0, A_{i,*}PSI⟨n_A⟩}}
Vector<n_G> e^{11} = E^{11}_{i,*}PSI⟨n_S⟩
Vector<n_G> e^{12} = E^{12}_{i,*}PSI⟨n_S⟩
Vector<n_G> e^{22} = E^{22}_{i,*}PSI⟨n_S⟩
Vector<n_G> P  = P^⋆ · h * exp(−20(1−a))
Vector<n_G> D  = (Δ²_min + (5/4)(E^{11}_{i,*}*E^{11}_{i,*} + E^{22}_{i,*}*E^{22}_{i,*}) + (3/2)E^{11}_{i,*}*E^{22}_{i,*} + E^{12}_{i,*}*E^{12}_{i,*})^{1/2}
Vector<n_G> P_D = P/D
S^{11}_{i,*} = (1−α^{-1})S^{11}_{i,*} + α^{-1}M_i^{-1}(P_D*((5/8)e^{11} + (3/8)e^{22}) − (1/2)P)
S^{12}_{i,*} = (1−α^{-1})S^{12}_{i,*} + α^{-1}M_i^{-1}(P_D*(1/4)e^{12})
S^{22}_{i,*} = (1−α^{-1})S^{22}_{i,*} + α^{-1}M_i^{-1}(P_D*((5/8)e^{22} + (3/8)e^{11}) − (1/2)P)
}
}
```

**Listing 1.** Implementation of the mEVP iteration Eq. (3). Stress and strain tensor components $S^{11}, S^{12}, S^{22}, E^{11}, E^{12}, E^{22} \in \mathbb{R}^{N \times n_S}$ are stored as matrices where $N$ is the number of elements and $n_S$ the number of local DOFs in the stress space. Ice height and concentration are denoted as $H, A \in \mathbb{R}^{N \times n_A}$, where $n_A$ is the number of local DOFs in the advection space. By $H_{i,*} \in \mathbb{R}^{n_A}$ (and similar for the stress and the strain) we denote the local row vector of the DOFs belonging to element $i$. The matrices $\mathrm{PSI}\langle n_A \rangle \in \mathbb{R}^{n_A \times n_G}$ are given at compile time and they evaluate the dG functions in the Gauss points with $n_G$ being the number of Gauss points. The scalars $P^\star, \Delta_{\min} \in \mathbb{R}$ are physical parameters and constant for a simulation. The matrices $M_i^{-1} \in \mathbb{R}^{n_S \times n_G}$ are pre-assembled and stored for each element. They represent the local inverse mass matrix scaled with the weights coming from the transformation of the mesh elements and multiplied with the matrix $\mathrm{PSI}\langle n_S \rangle \in \mathbb{R}^{n_S \times n_G}$. By "$*$" we denote the element-wise Hadamard product of matrices.

otherwise stated all numerical testcases use double precision. The code ,however, is generic in this respect and Section 4.2 will study the effect of using lower or mixed precision arithmetics.

For readers unfamiliar with GPU architecture and the specifics of GPU programming, we provide a brief introduction in Appendix A to complement the following text.

## 3.1   CUDA

The standard for general purpose GPU programming is CUDA (NVIDIA, 2023a), a platform developed by NVIDIA. The primary interface is a C based language and API with extensive compiler support for C++. CUDA has a mature ecosystem

and gives low level access to the GPU, which allows one to develop highly optimized code. However, CUDA is limited to NVIDIA hardware and the development effort to obtain code with a high utilization of the available compute resources can be considerable.

Since version 3.3, Eigen has limited support for CUDA and allows one to use fixed sized matrices in CUDA kernels. Through this, we can use the code from Listing 1 largely unchanged. Eigen's manually vectorized code paths need to be disabled to have
150 the code run on the GPU, but we still benefit from Eigen's other features such as expression templates and optimizations when a size is known at compile time. For the use of CUDA with Eigen, we have to ensure that the required data is in GPU memory. For dynamic buffers like $S^{11}$ in Listing 1, we allocate memory manually and copy data as needed before and after the kernel invocation. Inside the CUDA kernel, an Eigen::Map is constructed with

```
1       auto B = Map<Matrix<T, Dynamic,n>>(bufDevice, N, n) .
```

This provides the same interface as the original matrix. For compile time matrices such as PSI we use the GPU's *constant memory*. Advantages of constant memory are that no manual memory management is required, that memory access is faster through a dedicated cache, and that further compiler optimizations are possible since the values are available at compile time. In the original C++ CPU code, the constant matrices are defined as static class members with explicit template specialization to enable selection of the proper matrix for the specified dG-degree at compile time. Since static member variables are not
supported in CUDA, we instead declare separate variables and utilize **if** constexpr to achieve the same flexibility:

```
__constant__ constexpr T PSI_1_1[1] = {1.0};
template<int n, int nG> __device__ auto PSI(){
if constexpr (n == 1 && nG == 1) {
return Map<const Matrix<T,1,1>>(PSI_1_1);
}
}
```

Another important modification for the use of Eigen on the GPU is the use of 32-bit integers as index type, since the default 64-bit integers are only emulated on the GPU.

We tried a number of optimizations to speed up the Eigen CUDA code, the results of which are shown in Table 2. The
170 bottleneck on the GPU is often memory access. One remedy is the manual use of the L1-cache, called *shared memory* in CUDA. Shared between all threads in a thread block, it can significantly speed up reads of data that is needed multiple times and by multiple threads or when scattered memory reads/writes are necessary. In Listing 1, the only data that are used multiple times and by multiple threads are the PSI matrices. Only minor changes to the code are needed to load the PSI matrices into shared memory before use. However, we see no benefit from this change, cf. Table 2, since constant cache is just as fast for
the compile time matrices. Shared memory would therefore only be worthwhile if we expect to run out of constant memory. However, the size of the compile time matrices depends only on the local degrees of freedom of the discretization. If we consider all of the implemented discretization orders together, roughly 26 KB of memory are needed. This is still less than half of the 64 KB constant memory available (NVIDIA, 2023a).

| optimization | time [s] | speedup |
|:---:|:---:|:---:|
| CUDA baseline | $0.366 \pm 0.004$ | 1.0 |
| CUDA shared memory | $0.370 \pm 0.002$ | 0.99 |
| CUDA column-major | $0.419 \pm 0.002$ | 0.87 |
| CUDA on-the-fly map | $0.321 \pm 0.002$ | 1.14 |
| SYCL-AdaptiveCPP baseline | $0.466 \pm 0.002$ | 1.0 |
| SYCL-AdaptiveCPP shared memory | $0.532 \pm 0.001$ | 0.88 |
| SYCL-AdaptiveCPP on-the-fly map | $0.372 \pm 0.002$ | 1.25 |
| Kokkos baseline | $0.522 \pm 0.001$ | 1.0 |
| Kokkos shared memory | $0.551 \pm 0.002$ | 0.95 |
| Kokkos on-the-fly map | $0.386 \pm 0.002$ | 1.35 |

**Table 2.** Total time spend on the stress computation for $2.6 \times 10^5$ elements over 30 time-steps for the different implementations on an A100 GPU. Each modification is tested independently and speedup is relative to the respective baseline.

Another potential avenue to accelerate memory accesses is to carefully prepare the layout of the data. For the C++ CPU code, variables such as $S^{11}$ are stored in row-major order, meaning that coefficients belonging to the same cell are contiguous in memory. This locality is beneficial both for effective cache usage and for vectorized memory accesses. On the GPU, the most efficient way to access global memory is through *coalesced* reads whereby neighboring threads access neighboring addresses. Since each thread processes one cell, this can be achieved by storing variables in column-major order. Nonetheless, as we can see in Table 2, the switch to column-major storage order leads to a measurable slowdown. A profiler revealed that the use of a column-major layout does improve the memory access patterns and the number of excessive sectors loaded from global memory decrease from 59 % for the row-major version to just 2 %. However, this difference is rendered ineffective by the cache. In particular, data that is seemingly loaded without need in the row-major version due to the strided access is, in fact, required by subsequent computations when it can be read from the cache. Furthermore, the column-major version performs more instructions for index computations, leading to the overall slowdown.

A third option to reduce global memory accesses is to trade off reads with more computations. This is beneficial when the code is memory-bound, as is often the case on the GPU, especially with classical linear algebra (Dublish et al., 2017). In our code, the I/O can be reduced by re-computing the inverse parametric map $M^{-1}$, which depends only on the geometry of the mesh and compile time constants. In particular, each matrix has a size of $n_S \times n_G$ while each mesh cell's geometry is fully described by 4 vertices with 2 values each, which are furthermore shared with neighboring cells. So, disregarding constants, even for a small dG-degree such as $n_S = 3$, fewer reads are required if we compute the matrices on-the-fly, see also Richter et al. (2023a, Sect. 5.3.3). Upon closer inspection, we also find that when stored in column-major order, vertex reads are coalesced while reads to $M^{-1}$ are not, due to the fact that $M^{-1}$ is implemented as an array of matrices. Since the Eigen matrix type only deals with two dimensions, adjusting the storage order of $M^{-1}$ to allow for coalesced accesses would be difficult.

We find that the on-the-fly computation of $M^{-1}$ indeed delivers a speedup of $14\%$ over the CUDA baseline on an NVIDIA A100 GPU.

The above optimizations illustrate that GPU performance remains hard to predict and that for low level GPU programming, proper profiling is essential to identify bottlenecks and to develop efficient code. This applies not only to optimizations that address well-known bottlenecks, as above, but also to work on inconspicuous details in the code such as the order of expressions. An illustrative example of this is, again, found in the treatment of precomputed $M^{-1}$; this is relevant in particular for higher order discretizations that will be examined in greater detail in Sect. 4.3. Accessing the matrix by reference or by making an explicit copy has no impact on the performance for smaller matrix sizes like $3 \times 4$. However, when $M^{-1}$ has size $8 \times 9$, i.e. in a second order discretization, the copy results in a kernel that is $43\%$ faster overall. Curiously, the slowdown with the access by reference is largely not caused by a memory bandwidth bottleneck. Instead, the massive number of unique memory accesses overwhelms the instruction queue that is responsible for executing cached memory accesses. The copy of $M^{-1}$ alleviates this by encouraging the use of more registers to store the coefficients, thereby avoiding memory accesses.

## 3.2 OpenACC and OpenMP

A simple approach for moving computations to the GPU is to use a directive based programming model like OpenACC or OpenMP. In this case, only small or no changes to the code are required. Targeting C, C++ and Fortran, both OpenACC or OpenMP define directives to annotate loops. These instruct the compiler to offload the computations onto the GPU.

OpenACC or OpenMP differ in how the parallel execution is described. OpenMP is *prescriptive*, meaning that the programmer has to detail how a loop should be parallelized. On the other hand, OpenACC provides a simpler *descriptive* directive that leaves more decisions to the compiler. See (Usha et al., 2020) for more details on the differences between both approaches. In practice, OpenACC tends to give better performance (Usha et al., 2020; Đukić and Mišić, 2023). However, it has more limited compiler support. Except for basic support in GCC, OpenACC can only be used with experimental and commercial compilers that primary target NVIDIA hardware. Therefore, efforts exist to automatically translate OpenACC to OpenMP to access the larger ecosystem of OpenMP (Denny et al., 2018; Servat et al., 2022).

To accelerate our code, we tried three different compilers: GCC-12.2 and NVIDIA HPC-23.5 with support for both OpenMP and OpenACC, and Clang-16.0 which currently only supports OpenMP. However, we found that all three compilers fail for our code. The NVIDIA compiler refused to compile Eigen code, while GCC and Clang either crashed during compilation or produced a broken program that would crash once executed. Runtime crashes can be attributed to incorrect memory transfers, for which only Clang provided some diagnostics in the form of compile-time warnings. In particular, objects which are not trivially copy-able, such as Eigen matrices with at least one dynamic dimension, are not captured properly. While directives are provided to manually specify the needed buffers, this is cumbersome to do for the complicated template-based Eigen types in our code. Furthermore, it voids the main advantage of the directive based approach, namely its simplicity. Use of OpenMP and OpenACC was therefore not pursued further.

## 3.3 SYCL

SYCL (The Khronos Group, 2023a) is an open standard for heterogeneous computing developed by the Khronos group. The standard proposes a high-level API extending C++17 that allows the same code to run on various devices such as CPUs, GPUs and FPGAs. There are currently two major implementations of the SYCL standard, both of which are open source and build on LLVM. Development of AdaptiveCPP (Alpay and Heuveline, 2023), previously known as hipSYCL and OpenSYCL, is lead by Heidelberg University. While various backends are available, the focus is on NVIDIA and AMD GPUs. The other major implementation of SYCL is Data Parallel C++ (DPC++), which developed by Intel. DPC++ primarily targets Intel CPUs, GPUs and FPGAs.

SYCL builds on top of standard C++ to minimize the effort of adapting existing code. However, the SYCL standard forbids recursion and function pointers in kernel code (The Khronos Group, 2023b), both of which are used in Eigen's expression templates. DPC++ does not allow one to compile the neXtSIM-DG code because of these limitations, although the function calls should be entirely inlined in the compiled code. AdaptiveCPP requires more effort for setup but the tool chain compiles Eigen. We therefore limit our investigations to AdaptiveCPP in the following.

SYCL automates device memory management and movement of data between host and device, but memory requirements of a kernel need to be declared explicitly. To this end, a *buffer* needs to be defined, pointing to already allocated memory on the host. Then, a *command group* is created which collects all information needed to run a task in parallel. Inside the command group, *accessors* allow us to explicitly describe which buffers need to be accessed and how, i.e. read or write. Once pushed into a *queue*, the SYCL runtime uses these memory requirements as well as optional dependencies on other command groups to select the best suited memory region to perform needed memory transfers and to schedule the execution. Inside the command group, we can declare a parallel for-loop and construct Eigen maps analogous to CUDA with pointers provided by the accessors.

We can investigate the same optimizations as with the CUDA code. While shared memory did not improve performance for native CUDA, it is still of interest to see how it affects the SYCL implementation, since memory management works differently there. To access *local memory* in SYCL, which is the name used for CUDA's shared memory, we have to declare a *local_accessor* in the command buffer. In addition, local memory only makes sense in the context of thread blocks, so we need to use a more complicated for-loop which makes thread blocks explicit. Unfortunately, such a construct is known to perform far worse on the CPU than a simple loop and work on reducing this gap is an active area of research (Meyer et al., 2023). Therefore, if the code is to be efficient both on CPU and GPU, local memory should be introduced only in code paths specialized for the GPU. For the code snippet under study, this additional effort was not considered worthwhile. Returning to Table 2 we see using shared memory makes the kernel moderately slower. On the other hand, computing $M^{-1}$ on-the-fly leads to a more substantial relative speedup over the AdaptiveCPP baseline than the same optimization in native CUDA.

## 3.4 Kokkos

Kokkos (Trott et al., 2022) is another programming model to enable heterogeneous computing in modern C++, currently with support for CPU as well as NVIDIA and AMD GPUs. Kokkos is developed as part of the Exascale Computing Project by the US Department of Energy. The main difference to SYCL is that Kokkos is a library while SYCL requires compiler integration. The library-based approach greatly simplifies deployment of projects using Kokkos but potentially limits possible optimizations and features.

Kokkos consists of macros and wrappers that provide a unified API for the different backends with the final code being processed by the chosen compiler. Therefore, we can once again start from the CPU code shown in Listing 1, knowing that it works in native CUDA. The primary mechanism to manage memory in Kokkos are *Views*, which are basically a shared pointer to a multi-dimensional array. Typically, both a device view and a mirrored host view are created to facilitate data transfers. For our use case, it is possible to create a view on already allocated memory with the *unmanaged* trait. However, unmanaged views do not play well together with the mirrored views concept in backend agnostic code, leading to unnecessary copies during the execution on the CPU. Since in general the device view needs its own buffer, copies between the mirrored views will be performed regardless of whether they already reside in the same memory space. These extra copies can be avoided by adding a special case for just the view creation on the CPU. They are therefore not a major problem for portability. Once properly setup, data is accessible in the kernel through the device view and we can use the underlying pointer to create an Eigen map in the same manner as in CUDA.

Possible code optimizations in Kokkos are similar to those available in SYCL. CUDA's shared memory, called *scratch memory* in Kokkos, can be accessed by specifying a *TeamPolicy* with a thread block size instead of using a simple parallel for-loop. Here a nuisance of the library becomes apparent as the total scratch memory needed for a particular kernel has to be set manually. Furthermore, parallelism described with explicit thread blocks has the same downside as in SYCL, namely that it leads to strongly degraded CPU performance. In our tests we find that usage of scratch memory introduces a small overhead in Kokkos, see Table 2. On-the-fly map computation is again beneficial and it results in a large speedup of 35 %.

## 3.5 PyTorch

PyTorch (Paszke et al., 2019) is one of the most popular libraries for machine learning (Aoun et al., 2022). It consists of a simple-to-use Python frontend and a high-performance C++ backend that has a dedicated compiler to optimize code execution and maps execution for different hardware such as CPUs, GPUs, and TPUs. Full support is available for CPUs, NVIDIA GPUs and AMD GPUs.

To make effective use of PyTorch and the optimizations it implements, computations have to be reformulated in terms of large tensors. For this, we remove the main loop in Line 4 of Listing 1 and treat the element dimension $N$ as the batch dimension of variable size. The matrix-vector products then become matrix-matrix products and element-wise operations remain unchanged. Some care is necessary to perform the products with the per-element inverse maps, e.g. Line 16. Since we have a third dimension in $M^{-1}$, this is not a standard matrix-matrix product. However, we can map this operation to a batched

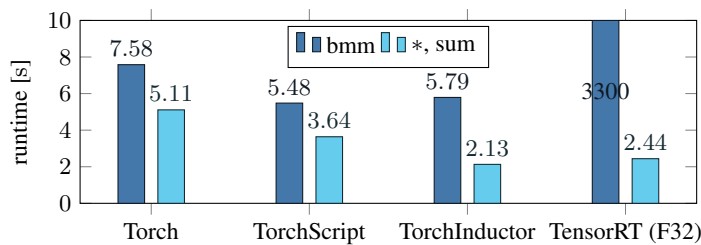

**Figure 1.** Total time spend on the stress computation over 30 time-steps for the different PyTorch variants on an A100. The products with $M^{-1}$ are implemented either as batched matrix-matrix product (bmm) or element-wise product and sum (∗,sum). TensorRT uses single precision (F32) since double is not supported.

matrix-matrix product (bmm) by appending a dimension of size 1 to the second argument and removing it again afterward (squeezing in PyTorch terminology). Alternatively, we can formulate this computation as an element-wise product by adding a dimension corresponding to $n_S$ to the second argument, followed by a sum over that dimension. The latter operation turns out to be 4 to 5 times faster across different backends, indicating that PyTorch is not tuned for our use case where the matrices are much smaller than those is common in machine learning workloads.

To integrate the PyTorch code into our C++ simulation, we have multiple options. With minor syntactic changes compared to the Python version, we can implement the computations directly with PyTorch's C++ API. However, this is inefficient since each operation is executed as a separate kernel with no kernel-fusion taking place, resulting in many reads and writes of the same data. A second option is to define the computation as a PyTorch model in Python. This model can be exported as *Torch-Script* and loaded in C++. Part of the C++ runtime is a just-in-time compiler which attempts to optimize the model execution on repeated use. However, more recent efforts to accelerate PyTorch models have been focused on *TorchDynamo*, a compiler first released with PyTorch 2.0. While the front-end of TorchDynamo is written in Python, various backends are available, some of which can be used without the Python runtime. Most promising among those we tested is the built-in *TorchInductor* which leverages the compiler Triton (Tillet et al., 2019) to produce highly optimized fused-matrix multiplications (PyTorch-devs, 2023). In particular, PyTorch 2.2 introduces AOTInductor, a version of TorchInductor that exports the entire model as a shared library with a single wrapper function to call directly from C++. Another way to deploy the PyTorch model in C++ is through the extension Torch-TensorRT (Torch-TensorRT-devs, 2024), which uses NVIDIA's inference engine TensorRT (NVIDIA, 2023b) as backend. One limitation of TensorRT is that it does not support double precision.

We compare the four proposed variants to integrate the PyTorch model into C++ in Fig. 1. Although they use the same tensor primitives, the native C++ interface is considerably slower than TorchScript. The new compiler, TorchInductor, with its Triton-optimized kernels, is significantly faster than the alternatives when using the element-wise product. When implemented with bmm, the compiler fails to optimize the operation due to a lack of GPU memory. The fact that a batch size of $2.6 \times 10^5$ is already too large, although the operands require less than 5 MB of memory, points to it being an edge case not properly considered by the optimizer. TensorRT is slower than TorchInductor for our use case, even while running in lower precision

and has even more trouble with the bmm operation. Optimization of the model takes over a minute and the inference is orders of magnitude slower than the sum-based version.

### 3.6 Development and deployment effort

The development of dedicated CUDA code is time-consuming and error-prone. One purpose of the alternatives we considered in this work is to reduce this high development effort. Furthermore, most of them support a unified code for a variety of compute hardware. We therefore do a qualitative comparison of the different approaches, considering ease of development but also deployment of the finished code on a target system.

With their modern C++ interface, both Kokkos and SYCL make it easier to write correct code compared to CUDA. Simplified resource management and stricter types reduce the risk of memory related bugs and make more errors visible at compile time. The simple parallelism constructs also hide GPU specific scheduling based on blocks and grids from a developer. A further advantage of SYCL is that explicit annotations of device functions are unnecessary. SYCL's memory model fully automates transfers between host and device, eliminating another source of errors. It should be noted, however, that the more advanced C++ features used by Kokkos and SYCL can make the frameworks less approachable for non-C++ experts than the C-like interface of CUDA. PyTorch follows a completely different programming paradigm from the other options. From a system programming language perspective, PyTorch takes time to get used to. Development in PyTorch is, however, overall much simpler. There is no memory management or explicit parallelism to take care of and rapid prototyping in Python is possible. A potential downside can be that some computations are hard to express in terms of tensor operations, in which case a low-level, manual implementation is still needed. Another downside for our particular case was that the code had to be completely rewritten in PyTorch, while for the other options the C++ CPU code could be largely reused.

For running the code on a target system, pure CUDA is easiest. Usually pre-installed on clusters, no additional setup is required. Furthermore, CUDA (or the AMD equivalent ROCm) is a prerequisite for the other frameworks to use the GPU, so if a manual installation is needed, this effort is unavoidable for every framework. The Kokkos library can be easily integrated into a project's CMake based build system and then works out-of-the-box. In combination with automatic fetching, e.g. via git submodule, the library setup becomes transparent. SYCL requires a specialized toolchain. For AdaptiveCPP, this means that its compiler wrapper has to be first build from source. The manual configuration that is needed for AdaptiveCPP to find the proper compilers is cumbersome and, in some cases, necessitate building a suitable version of LLVM first. For PyTorch, prebuild C++ libraries are available for all supported platforms. To use TorchInductor, the Python package is needed to generate the code on the target system, but it is easily acquired through a package manager.

## 4 Numerical experiments

To analyze the performance of our implementations, we use the established VP benchmark of a moving cyclone over a sea ice region (Mehlmann et al., 2021). We simulate a duration of $1 \, \mathrm{h}$, which requires 30 advection steps and $3,000$ stress updates. For the discretization we choose first order continuous Galerkin (cG) elements and discontinuous Galerkin (dG) elements with 3

| | System 1 | System 2 |
|---|---|---|
| CPU | $2\times$ AMD EPYC Rome 7402, $2 \times 24$ cores @ 2.8 GHz | AMD EPYC 7A53, 64 cores @ 2.75 GHz |
| GPU | NVIDIA A100, 40 GB HBM2e | AMD Instinct MI250X, 128GB HBM2e |
| OpenMP (CPU) compiler | GCC-12.3 | - |
| GPU software stack | CUDA 12.2 | ROCm 5.6.1 |
| Kokkos | 4.1.0 | 4.1.0 |
| AdaptiveCPP | 23.10.0 based on Clang-17.04 | 23.10.0 based on AMD Clang-16.0 |
| PyTorch | 2.3 Nightly (24-November-2023) | 2.4 (ROCm 6.1.2) |

**Table 3.** The two systems on which performance measurements where conducted.

degrees of freedom for the advection. The original C++ CPU version of the code has already been validated on this benchmark, see (Richter et al., 2023a). We therefore compare to the CPU version to validate the computed results and note that, while there is no loss in accuracy between the implementations, deviations from double machine precision can cause visible differences in the results over longer time-scales. See Sect. 4.2 for additional details.

In the GPU implementations, significant time is required to transfer memory between host and device. Nonetheless, we only consider the kernel execution times in the following since the final objective of our work is a full GPU implementation of the dynamical core. While transfers are still necessary for coupling with other models, the major effort of simulating the sea-ice dynamics is in the mEVP iteration with the many sub-iterations considered in this work. This will amortize the costs of the memory transfers. Furthermore, they can potentially be hidden by overlapping them with computations. To ensure accurate

timings, synchronization barriers are inserted as needed before and after the kernel invocation. In SYCL, memory transfers are implicit, so we rely on the built-in profiling instead to obtain the timings for SYCL-AdaptiveCPP. Details on the software and hardware used in the experiments are listed in Table 3.

## 4.1 Performance scaling

Of particular importance for coupled climate simulations is the scaling of the performance as a function of grid resolution.

With a fixed domain size of $512$ km, we reduce the cell size from $4$ km to $0.25$ km. This corresponds to an increase in the number of elements from $1.6 \times 10^4$ to $1.7 \times 10^7$ (i.e. one has a quadratic scaling of the element number in the resolution). In Fig. 2 we compare the best implementation for each approach as a function of elements for two different data center GPUs.

    For the OpenMP reference we obtain the best results by utilizing all available threads with simultaneous multithreading enabled and the settings `OMP_PROC_BIND=spread` and `OMP_PLACES=threads`. In the GPU versions of CUDA, Kokkos

and SYCL-AdaptiveCPP, we compute the inverse maps on-the-fly. In case of Kokkos and SYCL-AdaptiveCPP, the simple for-loop is used to run the update in parallel, which also makes an execution on the CPU efficient, albeit with precomputed maps. For Kokkos on the CPU, we get the best results with the OpenMP backend and the same settings as raw OpenMP. For PyTorch, we take the implementation generated by TorchInductor with the element-wise product and sum.

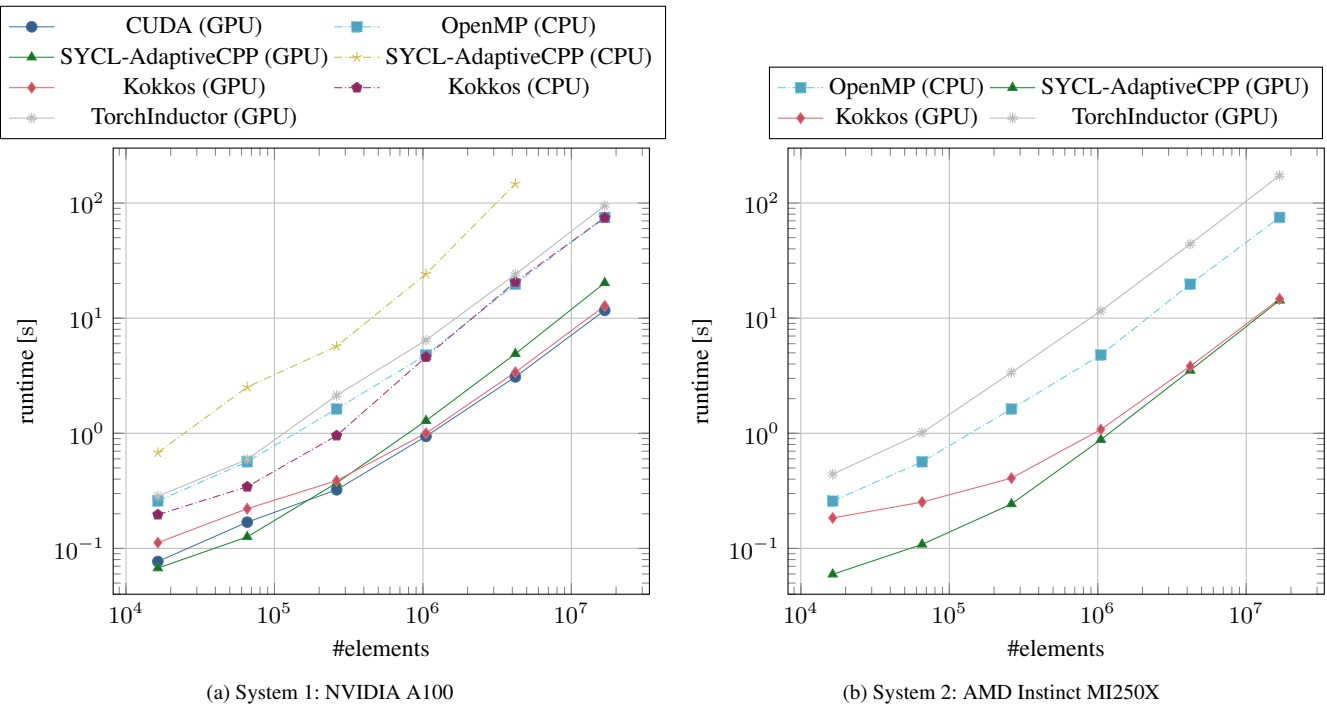

(a) System 1: NVIDIA A100        (b) System 2: AMD Instinct MI250X

**Figure 2.** Timings of the stress update using the best performing version for each framework. The size of the mesh cells size is scaled from 4 km to 0.25 km while keeping the domain size constant to increase the number of elements. On the AMD Instinct MI250X, only one of the two graphics compute dies (GCDs) is used. The values for OpenMP (CPU) are the same in both plots and where measured on System 1 with 96 threads.

Running on a NVIDIA A100 GPU (Fig. 2a), our CUDA implementation delivers a significant speedup over the OpenMP

CPU reference implementation. For the smallest problem with $1.6 \times 10^4$ elements, CUDA is 3 times faster, scaling up to $6.4$ for $1.7 \times 10^7$ elements. Kokkos achieves asymptotically the same performance as CUDA on the GPU and as OpenMP on the CPU. This is to be expected since the very same compilers (NVCC and GCC) are used by Kokkos and only memory buffers and kernel dispatch are abstracted. On small problems, Kokkos overhead makes it 50 % slower than CUDA but, surprisingly, the CPU version is slightly faster than raw OpenMP for the same number of elements. SYCL-AdaptiveCPP scales worse than

the other GPU accelerated codes, being 70 % slower than CUDA for the largest problem size we tested. However, it still provides a significant improvement over the CPU OpenMP version running with 96 threads. The good performance of SYCL-AdaptiveCPP for small problem sizes is likely an artifact from the different time measuring method, since the SYCL timings do not fully account for the kernel launch overhead. However, since this cost does not scale with the number of elements, it becomes insignificant for larger problems where the GPU implementations are most useful. On the CPU, we were not able to

run a meaningful experiment with SYCL-AdaptiveCPP. Best performance was achieved with a restriction to just 24 threads, indicating that the available CPUs are not utilized properly. The documentation states that performance of the CPU backend should be similar to raw OpenMP and that a significant deviation is likely caused by an improperly configured toolchain.

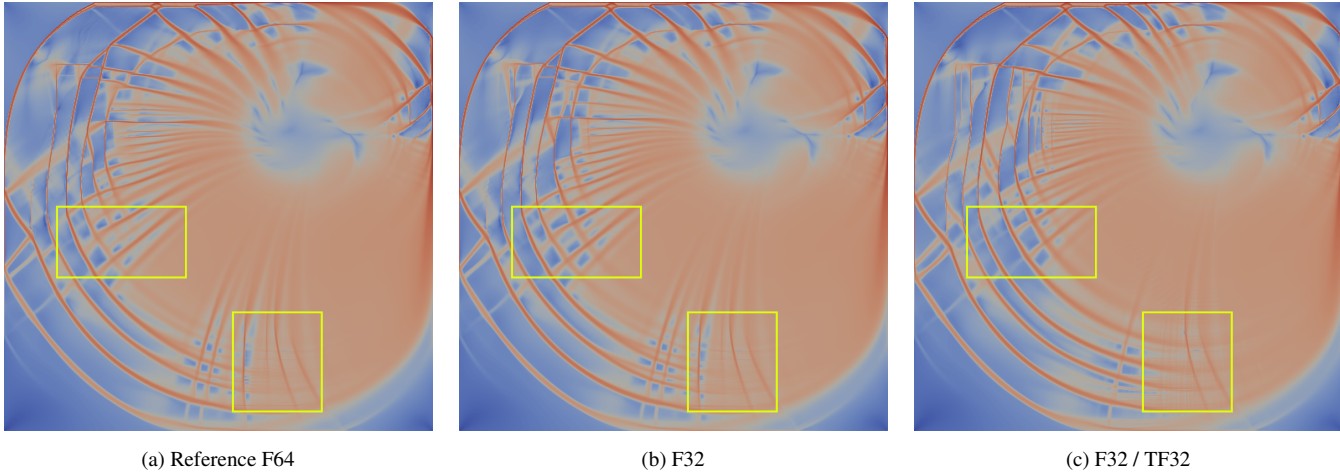

|     |     |     |
| :-: | :-: | :-: |
| (a) Reference F64 | (b) F32 | (c) F32 / TF32 |

**Figure 3.** Shear deformation ($\log_{10}$) for the benchmark after $48$ h where the stress update is computed with different floating point types.

However, we were unable to obtain stable results on three different systems, illustrating the substantially greater difficulty in using the framework compared to Kokkos. TorchInductor on GPU is slower than the OpenMP CPU code for every problem size tested and TorchInductor's CPU code (not shown) is an order of magnitude slower than the GPU version.

To test the portability of the heterogeneous compute frameworks we also ran the experiments on System 2, equipped with an AMD MI250X GPU. The results are shown in Fig. 2b. Kokkos, SYCL-AdaptiveCPP and PyTorch work without modifications but only utilize half of the MI250X, since it is a dual graphics compute die (GCD) design. SYCL-AdaptiveCPP performs better on the AMD GPU, while Kokkos is somewhat slower than on the NVIDIA A100. Both thereby achieve a similar runtime, roughly $25$ % higher than that of CUDA running on the A100. For PyTorch, the MI250X is $50$ % slower than the A100 in our experiments. On paper, a single graphics compute die of the MI250X has the same memory bandwidth as the A100 and a F64 peak performance more than twice as high (A100 9.7 TFLOPS, MI250X 23.9 TFLOPS). These results indicate that the AMD ecosystem is still less mature. However, with the performance currently achieved it is still a worthwhile target platform.

### 4.2 Mixed Precision

One avenue to further speed up the simulation is to perform computations with lower precision float types. Switching from double precision (F64) to single precision (F32) halves the memory required and doubles the theoretical peak performance achievable on a A100 (F64 9.7 TFLOPS, F32 19.5 TFLOPS). Modern GPUs support even lower precision types that promise further speed-ups. Of particular interest is tensor float (TF32), a format used in tensor cores, which are specialized matrix multiplication hardware found on NVIDIA GPUs and originally introduced for machine learning workloads. TF32 uses the same exponent as F32, so that the range of representable numbers is the same as for F32, combined with a half precision mantissa with just $10$ bits that yields a significantly higher peak performance for matrix multiplications (156 TFLOPS) compared to F32.

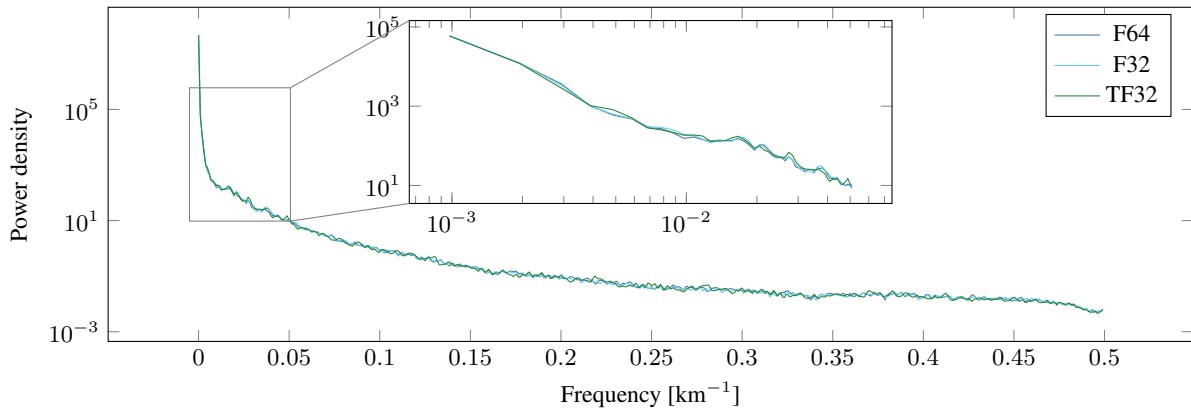

**Figure 4.** Radially averaged power spectral density (RAPSD) of the shear deformation ($\log_{10}$).

First we evaluate the impact of lower precision floats on the quality of the results. To this end, we perform the stress update in F32 while the rest of the simulation still runs in F64. To make use of the tensor cores, we run the PyTorch version, where TF32 can be enabled with a simple switch. Results for the benchmark after $48\,\mathrm{h}$ simulation time are shown in Fig. 3. Visually, F64 and F32 are indistinguishable, while some movement of larger cracks and additional fine features can be seen with TF32. Nevertheless, the overall distribution of features in the shear deformation remains very similar and there is good agreement in the frequency spectrum of the features, as can be seen in Fig. 4.

To quantify the stability of the physical model itself and whether computations with lower accuracy amplify instabilities, we look at the distribution of linear kinematic features (LKFs) (Kwok, 2001) using a script provided by Hutter et al. (2019). In Fig. 5, we investigate the influence of randomly perturbed initial conditions on the formation of LKFs in the benchmark simulation. For F64 and F32, both mean and standard deviation are largely the same and perturbations up to F32 machine precision around $10^{-7}$ appear to have no impact on the distribution. This indicates that, at least for short term simulations, F32 can be used without impacting the results. For TF32, there is a statistically significant deviation in the length of LKFs even for very small perturbations. However, the uncertainty in the sea-ice tracers from data assimilation is of the order $10^{-2}$ or larger (Liu et al., 2019; Xie et al., 2017). In our experiment, the variation of the initial conditions has a larger effect on the LKFs than the floating point type used at this point, so TF32 could still be useful in practice.

The performance gains from switching the stress update to F32 or TF32 are shown in Fig. 6. For CUDA, switching from F64 to F32 gives a speedup of $80\,\%$, which is in line with the expected speedup for the A100, since both compute and memory throughput are effectively doubled. For this reason, we expect to see similar gains from using F32 in Kokkos and SYCL-AdaptiveCPP, albeit with the different compiler of the latter, other factors including occupancy and the use of special function units could play a larger role. For TorchInductor, the relative speedup with F32 over the F64 version is much higher at $315\,\%$. As machine learning tasks rarely use double precision, the optimizer is likely tuned much more for the single precision case. The variant with tensor cores enabled, TorchInductor (TF32), has a speedup of $412\,\%$ over F64 which is not quite as large as the theoretical peak performance would suggest. This is because not all operations can make use of the tensor cores and where

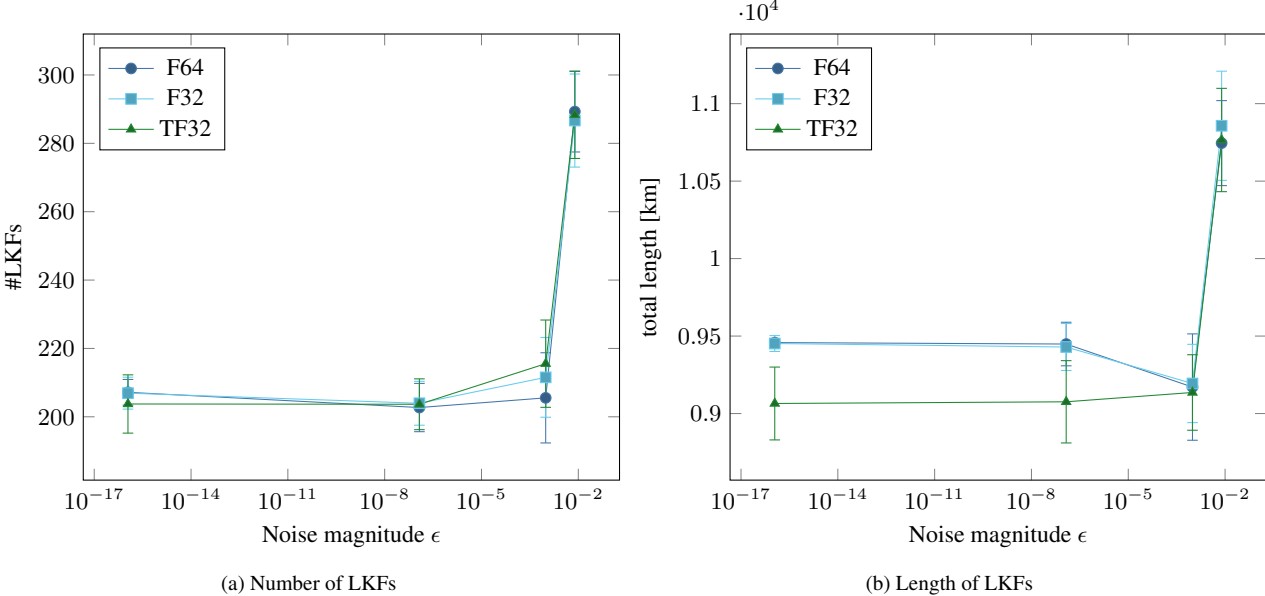

(a) Number of LKFs

(b) Length of LKFs

**Figure 5.** Distribution of linear kinematic features (LKFs) after $48$ h when the initial state of ice height $H$ and ice concentration $A$ are perturbed by adding uniform noise sampled from $[-\epsilon, \epsilon)$ to each component of the fields. Points show the mean of $24$ randomized runs with a specific maximum magnitude $\epsilon$, while the whiskers indicate the standard deviation. The $\epsilon$ values are chosen as the interval machine precision for different floating point types, i.e. $2^{-53}$ (F64), $2^{-23}$ (F32), $2^{-10}$ (F16/TF32), $2^{-7}$ (BF16).

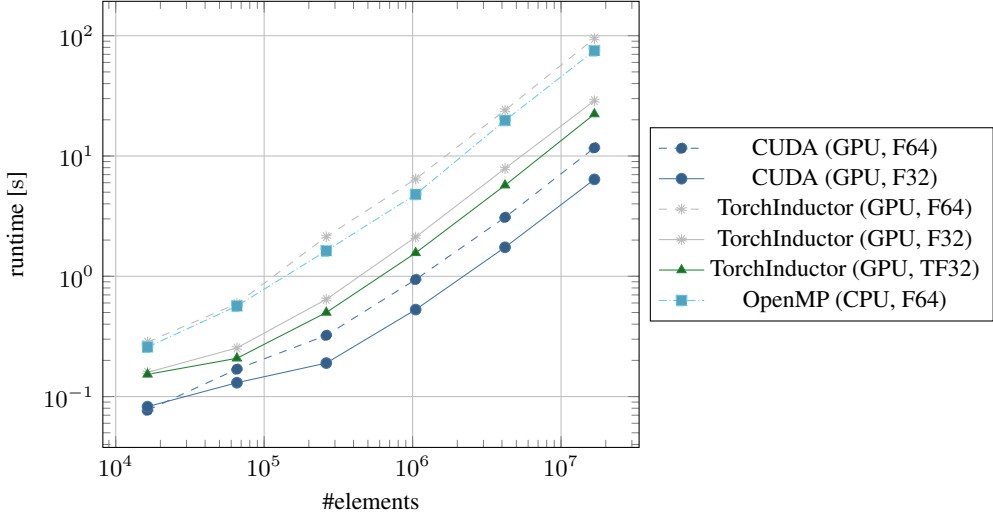

**Figure 6.** Timings of the stress update for lower precision floating point types on the NVIDIA A100. The dashed lines are references in F64, taken from Fig. 2.

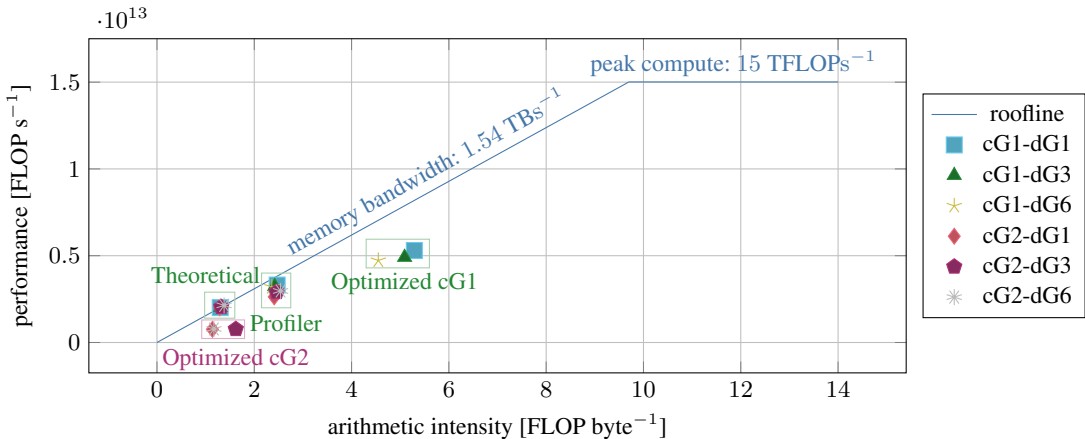

**Figure 7.** Roofline model of the F32 performance of the A100 with different variants of our CUDA kernel. Theoretical values are computed manually, while the other values are taken from Nsight Compute, the NVIDIA GPU profiler. The peak compute value given here assumes base clock speed to be consistent with the profiler results. Optimized kernels perform the on-the-fly map computation which is not viable for higher cG order.

it is possible, the matrices involved are too small to take full advantage of the tensor cores. In absolute terms, TorchInductor (TF32) still takes almost twice as long as CUDA (F64), even with access to tensor cores.

## 4.3 Higher order scaling

Our finite-element code makes it easily possible to change the local number of degrees of freedom. So far we have focused
on just one discretization, cG1-dG3, i.e. first order continuous Galerkin (cG) elements for velocity and 3 degrees of freedom discontinuous Galerkin (dG) for the advection. A higher order discretization significantly increases the compute load. This could make the computations more efficient on the GPU when compared to the CPU, since the GPU is limited by memory bandwidth in our code. On the A100, compute throughput reported by the profiler for cG1-dG3 varies from 28 % to 50 %, depending on whether the precomputed maps are used, while the memory throughput is at 70 % to 80 %.
A simple roofline model of the A100 GPU is shown in Fig. 7. For our code, the relevant bottleneck is clearly the *arithmetic intensity*, i.e. the ratio of compute operations per byte accessed in global memory. The values reported by the profiler for our baseline kernels with precomputed maps are all close to the bandwidth limit, which underscores our observations from Sect. 3, i.e. that the only effective optimization is a reduction of the data transfer per computation.

    In contrast, the optimized kernels with on-the-fly parametric map computation present a different picture where neither
memory bandwidth nor compute are a bottleneck. Instead, other limitations become apparent which can be attributed to the complexity of our kernel. One aspect that is exacerbated by the on-the-fly map computation, is the high number of registers that are needed by each thread. More temporary variables are needed and although, for the tested cG1 kernels, these variables still fit in the registers, the higher register usage leads to a lower *occupancy*, i.e. the ratio of active warps that fit on each

| cG \ dG | $n_A = 1$ | $n_A = 3$ | $n_A = 6$ |
|---|---|---|---|
| 1 ($n_S = 3, n_G = 4$) | 1.30 | 1.31 | 1.31 |
| 2 ($n_S = 8, n_G = 9$) | 1.29 | 1.32 | 1.38 |

**Table 4.** Estimated arithmetic intensity [FLOP byte$^{-1}$] of kernels with different discretization orders for F32. The roofline for the A100 is 9.73 FLOP byte$^{-1}$ (or 12.5 FLOP byte$^{-1}$ with boost clock). The arithmetic intensity depends on the chosen degrees of freedom $n_A$ in the dG advection space, as well as number of stress components $n_S$ and gauss points $n_G$ which are both determined by the cG order.

execution unit at the same time and the theoretical maximum supported by the hardware (for cG1-dG3: 57 % unoptimized, 40 % optimized). As a result, the scheduler is unable to keep the different pipelines busy and more cycles are wasted waiting.

While the gains in arithmetic intensity from the optimization are clearly worth it for cG1, cf. Table 2, the same is not true for cG2. With second order cG elements, the increased workload per thread, caused by the on-the-fly map computation, becomes a major bottleneck. The computation involves a matrix inverse that becomes both too expensive and memory intensive for higher orders. For first order cG, this matrix has size $3 \times 3$ and we can use a closed-form formula to compute the inverse that results in efficient code. For second order cG elements, the matrix has size $8 \times 8$ and we have to rely on a generic algorithm for matrix inversion. As we can see in Fig. 7, the arithmetic intensity of the "optimized" cG2 kernels is, in fact, lower than that of the kernels with precomputed maps because of a large number of temporary variables stored in global memory, since the available per-thread register space is exhausted (in technical terms data is *spilled* to *local memory*). Subsequently, the unoptimized kernel is roughly 8 times faster. In the following, we therefore limit our analysis to the version with precomputed maps.

Although the arithmetic intensity is reduced, the high order discretisation is still more powerful when it comes to resolving local features in sea ice. This is known from CPU implementations, see Mehlmann et al. (2021); Richter et al. (2023b) and Fig. 8 will demonstrate that the GPU efficiency does not depend significantly on the order.

To analyze the possible throughput independent of (sub-optimal) code generation, we also compute a theoretical bound based on the algorithmic description in Listing 1. For memory accesses we only count the data that is unique for each kernel invocation, i.e. $S_{i,*}, E_{i,*}, H_{i,*}, A_{i,*}$ and $M_i^{-1}$, giving us a total of $9n_S + 2n_A + n_S n_G$ float read and writes. For compute operations we only count float operations and assume F32, since in that case, all operations are native GPU instructions with well documented throughput (NVIDIA, 2023a). Additions are not counted since they can all be executed as fused-multiply-add, min and max have the same throughput as compute operations, and exp2 and sqrt$^{-1}$ count as 4 operations each (compute capability 8.0). Therefore, the total number of operations is

$$6n_S + 3n_A + 6n_S n_G + 2n_A n_G + 28n_G,$$

giving us the theoretical arithmetic intensities recorded in Table 4. We can see that, in agreement with the profiler results, an increase in either order is insignificant for the arithmetic intensity. The largest difference of just 6 % is from cG2-dG1 to cG2-dG6, which should not have much of an impact on the memory access bottleneck.

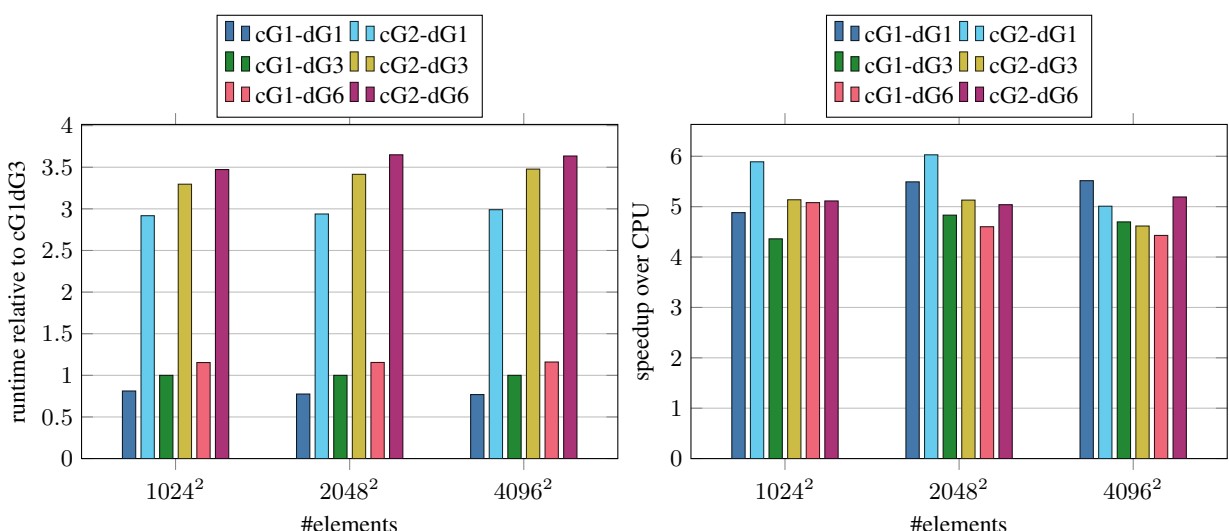

**Figure 8.** Performance of the CUDA implementation with precomputed map $M^{-1}$ for different discretization orders on the A100. The three largest problem sizes from the scaling test are shown.

To verify this scaling model we measure the runtime for different discretization orders and compare with the OpenMP
CPU baseline in Fig. 8. We find that the speedup from using the GPU ranges from $4.4$ to $6.0$ with no clear trend for either order parameter, inline with the theoretical prediction. The observed differences can mainly be attributed to the problem size impacting the behavior of the CPU version, since the relative runtime between the different GPU variants is very consistent across problem sizes.

## 5 NeXtSIM-DG implementation

Based on the results from Sect. 4 we decided on Kokkos for the full implementation of the neXtSIM-DG sea ice dynamical core. In our evaluation Kokkos offered performance almost competitive to CUDA but with greater ease of development, multi-vendor GPU support and the potential to replace the dedicated CPU implementation as well.

The port of the complete mEVP iteration to Kokkos is, for the most part, straightforward with the experience gained from the implementation of the stress. Of note is the need for a different parallelization strategy for operations that involve neighboring
cells of the mesh such as the divergence computation. In the OpenMP implementation, race-conditions are circumvented by performing the update in two steps, whereby every other row is processed in parallel (Richter et al., 2023a). However, as illustrated in Fig. 9, this parallelization strategy along just one dimension does not provide nearly enough work to saturate a GPU. One way to further increase parallelism without introducing contention at the edges is to perform four separate steps in a strided checkerboard pattern instead. Another way, that turns out to be faster for the divergence computation, is to process

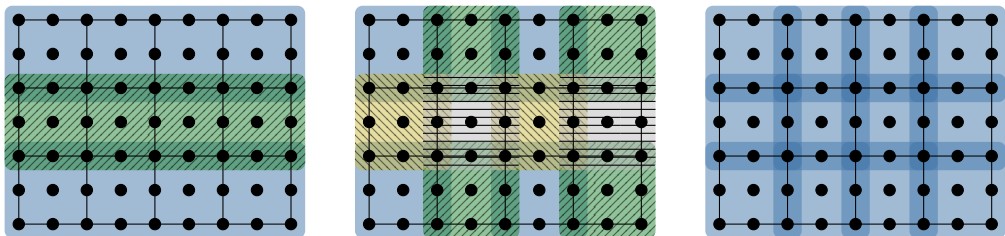

**Figure 9.** Parallel processing of biquadratic cG2-vectors. The mesh has 12 elements with $N_x = 4, N_y = 3$. Each block corresponds to one thread and blocks of the same color can be processed in parallel. The maximum number of parallel tasks is, from left to right: $\frac{N_y}{2}$ (row-wise), $\frac{N_x N_y}{4}$ (strided), $N_x N_y$ (atomic).

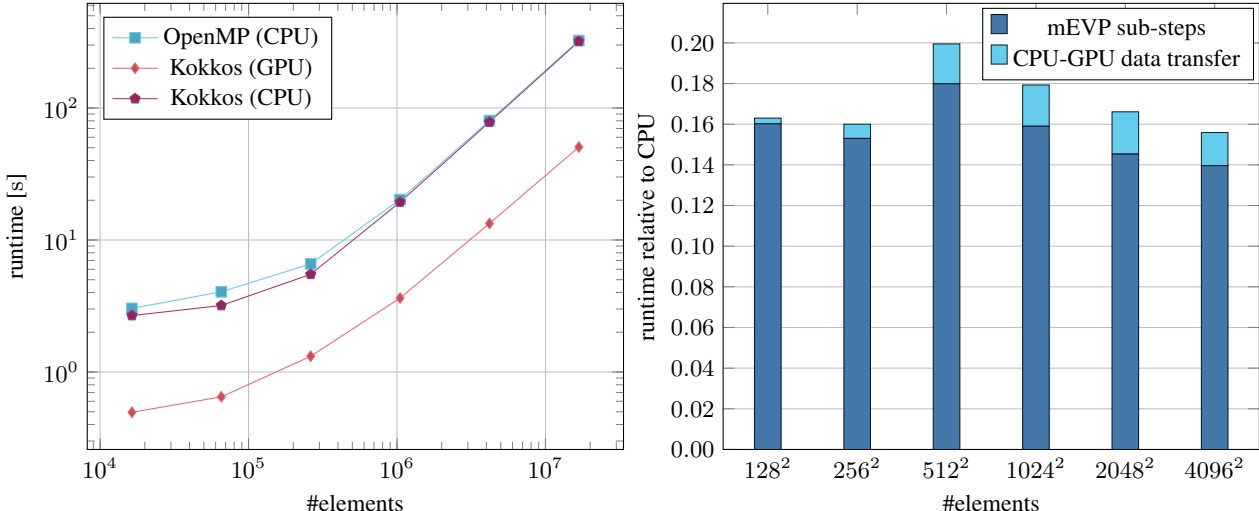

**Figure 10.** Timings of the full mEVP iteration on the NVIDIA A100 (System 1). The values for Kokkos (GPU) include the necessary data transfers between CPU and GPU. The CPU runs are with 96 threads.

every cell in parallel and to rely on atomic operations to ensure that the values are updated correctly. The use of atomics in such a way could be suboptimal when running on CPU, but it does not appear to be a problem in our experiments.

Another concern for portability, also related to cross-cell updates, is the choice of the parallel loop construct. To easily address neighbors it can be tempting to use Kokkos's 2D range policy. However, this policy splits work into tiles instead of rows, which does not play well with our underlying data layout on the CPU. The whole mEVP update becomes 3 times slower on the CPU when using 2D-loops for strain and divergence computations, instead of 1D-loops with manual index computation. GPU performance is, in our case, largely unaffected by this choice.

| order | base memory [MB] | per-element [KB] | max elements with 40 GB |
|-------|-----------------|------------------|--------------------------|
| cG1-dG3 | 463.5 | 0.70 | $5.6 \times 10^7$ |
| cG2-dG6 | 463.5 | 3.44 | $1.2 \times 10^7$ |

**Table 5.** GPU memory usage for different discretization orders. A constant amount of memory is reserved by the runtime. Memory requirements for data scale linearly with the number of mesh elements, giving us an upper limit for the problem size on a 40 GB GPU like the A100.

## 5.1 Benchmark

While the new Kokkos code is not fully optimized yet and limited to F64, it already provides a substantial speedup over the OpenMP code. Results for the benchmark setup described in Sect. 4 are presented in Fig. 10. In contrast to the previous experiments, the timings for the full mEVP iteration include the necessary data transfers to and from the device. Running on GPU, the mEVP iteration is faster by a factor of 6, even for small problem sizes. Running on CPU, the Kokkos code performs just as well as the OpenMP code. The cost of data transfers for the GPU version is still non-negligible at a constant 10 % for larger meshes, even though the 100 mEVP sub-steps are performed entirely on the device. This underlines the importance of minimizing necessary data transfers and running larger parts of a simulation on the GPU.

Another important aspect of the GPU implementation are the device memory requirements. Especially due to the precomputed matrices, the available device memory does impose a practical limit on the mesh size. The memory requirements are recorded in Table 5. Apart from a constant part that is mostly reserved for the runtime and which varies slightly between devices, the memory usage scales linearly with the number of elements. A GPU with 40 GB VRAM can therefore fit roughly $5.6 \times 10^7$ elements of order cG1-dG3. A switch to the higher order cG2-dG6 reduces the maximum number of elements to a little over one-fifth of that number.

As more parts of the simulation, such as the advection, are performed on the GPU, the memory requirements increase further. Possible strategies to reduce the device memory usage include moving buffers to RAM while they are not needed and to precompute only parts of the required maps to facilitate greater reuse between kernels. Both strategies are a trade-off between speed and memory. An alternative option is be to distribute the work across multiple devices. While this introduces synchronization overhead, the device memory usage itself should then not be a bottleneck as our results in Fig. 10 show that a single GPU can be well utilized with $10^6$ elements or more.

## 5.2 Realistic case

So far we have worked with an idealized test case in Cartesian coordinates that has a square domain which is completely covered by sea-ice except for a single boundary layer. In a realistic setting there will areas of land and open sea which do not need to be simulated. Although the parametric mesh can be deformed to closely fit a coast-line, the regular structure of the mesh will leave some computational cells inactive when more complicated geometry is involved. We therefore considered also

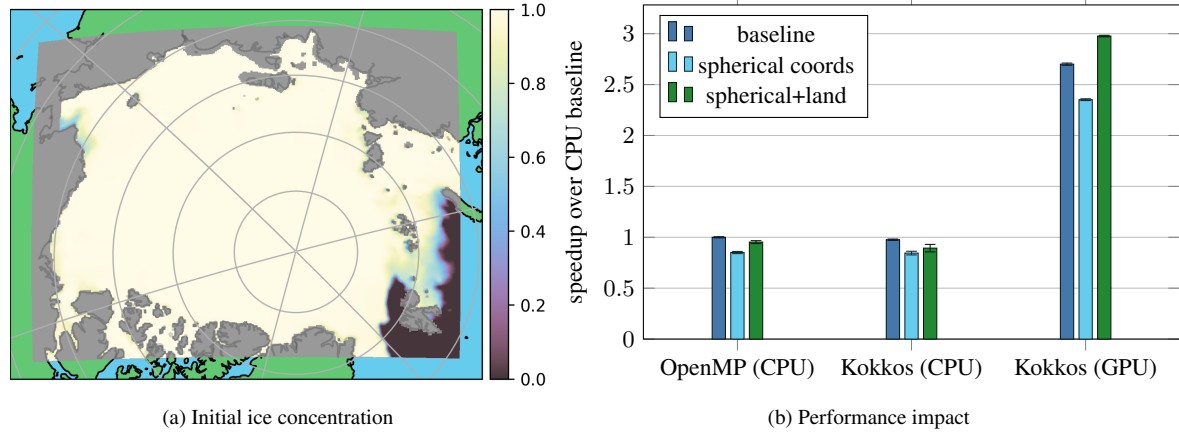

(a) Initial ice concentration

(b) Performance impact

**Figure 11.** A more realistic test case with a mesh of dimensions $1240 \times 980$ ( $1.2 \times 10^6$ elements) and resolution 3.125 km covering the arctic. The data is for the 01 January 2010 with initial conditions and ocean forcing derived from TOPAZ4 reanalysis (Sakov et al., 2012) and ERA5 reanalysis (Hersbach et al., 2020) for atmospheric forcing. Performance measurements are taken on a system with $2\times$ AMD EPYC 7773X (64 cores each), using 122 OpenMP threads in total and a NVIDIA H100 PCIe. Factors that affect the performance compared to the benchmark test case are the use of spherical coordinates and a land mask (gray) which leaves only 68 % of the cells active. Performance values shown are relative to the idealized CPU (OpenMP) baseline without land and with Cartesian coordinates. Whiskers indicate the standard deviation from 8 runs.

a more realistic setup in our experiment, see Fig. 11. Once again, the OpenMP reference and Kokkos (CPU) versions perform very similar. The GPU baseline is around 2.7 times faster than the CPU. The smaller performance gap compared to previous experiments can be explained by the differences in hardware. The CPUs are significantly faster and we use 122 physical cores instead of 48 as before. While the H100 that is used for the experiments is also faster than the A100 in the other experiments, the main bottleneck, the memory bandwidth, does not improve significantly form the A100 (1.6 TB) to the H100 PCIe 2 TB.

Spherical coordinates, which introduce additional scaling terms, lead to a performance penalty of around 15 % for all tested variants. The land mask, which deactivates updates for one-third of the domain, has little impact on the CPU. With a 12 % improvement for OpenMP and 6 % for Kokkos (CPU), the final measured times are slightly slower than the baseline. The small improvement from the land mask is because of the static parallel schedule, whereby each thread receives the same number of elements to update, leaving the thread with the most active elements to determine the overall speed. This will be addressed in the MPI version of the code, where the domain decomposition can take into account land cells. The GPU version, on the other hand, can already take advantage of the inactive cells to some extent, since work is split into many small warps, which finish instantly when they consist entirely of land. In fact, a speedup of 26 % from setting the land mask makes the GPU code faster overall for the realistic case than for the idealized baseline. To get this speedup, there needs to be larger areas of land and the mesh resolution has to be sufficiently high, so that the ratio of warps consisting of both active and inactive cells is small compared to the total.

The different characteristics regarding the land mask lead to the first platform specific adoption in our code. For kernels like the stress, which do not affect neighboring cells, land checks are optional and consequently inserted in the GPU version, but left out when compiling for the CPU. The presented values in Fig. 11 already include this optimization.

Finally, the memory usage does not change for the realistic case, so the values in Table 5 remain valid. The additional values used for spherical coordinates are currently loaded on to the device regardless of whether they are needed. The land mask has no effect since the space for land cells in the field arrays still has to be allocated to allow for direct access of the cell-data with computed indices.

## 6   Conclusions

We implemented and evaluated different options for the GPU parallelization of the neXtSIM-DG dynamical core. According to our results, CUDA remains the most reliable option both in terms of performance and with regard to the toolchain. Thanks to the CUDA support of Eigen, we were also able to use the CPU C++ code with minimal modifications in CUDA.

Kokkos benefits in the same way from Eigen's CUDA library support, while SYCL does not need explicit support which makes it well suited for an incremental port of existing C++ code in general. The streamlined memory model and simplified parallel constructs of Kokkos and SYCL facilitate more effective development, but at some performance cost. Using dedicated GPU features such as shared memory remains an issue because it leads to code that is very inefficient on the CPU, breaking the promise of the heterogeneous computing paradigm. However, our study demonstrates that this specialization is not always needed to achieve good performance and we can therefore recommend Kokkos as an alternative to CUDA. While SYCL shares the same benefits on paper, it suffers from immature implementations and is currently too unreliable for practical use.

PyTorch currently lacks far behind the more conventional options in terms of performance and is therefore mostly worth considering for rapid prototyping. However, the optimizer heuristics are clearly far from optimal yet for our use case and the underlying compilers are developing quickly, so we expect performance improvements in the future. Furthermore, PyTorch and similar machine learning frameworks are interesting alternatives because of their ease of development and the access to automatic differentiation they provide. The latter is of great relevance for hybrid methods that combine a conventional discretization with a machine learning component, e.g. Bedrunka et al. (2021); Kochkov et al. (2021); Demeure et al. (2023); Kochkov et al. (2023).

Our investigation of mixed precision underscores previous results from the literature, i.e. that lower precision float types should be considered for GPU codes. Performing a major computation of our sea ice simulation in single precision shows no degradation in the results while almost doubling the performance. Although the application of tensor cores with their even lower precision does have a measurable impact on the results, further tests with more realistic scenarios will be needed to determine whether there is a practical impact of going below single precision, and to study the effects of numerical precision on long-term climate simulations.

From a computational perspective, our finite element based GPU code does not favor higher order discretizations. While the speedup over CPU is considerable for all combinations tested, there are additional optimization opportunities for the first order

continuous elements which makes them more efficient. A comprehensive evaluation of the trade-off between quality and speed of different discretization orders is left for future work.

For the GPU-parallelization of the entire neXtSIM-DG dynamic core we chose Kokkos. For the full mEVP iteration running entirely on GPU we obtain a speedup by a factor of 6, switching from a dual CPU node to a single A100 GPU in double precision. With some care, the GPU Kokkos implementation achieves the same performance on CPU as our manual OpenMP based implementation, making the latter obsolete. The code is not fully optimized yet and more work is needed to port advection and other rheologies. With components outside the dynamical core also still in active development, performance comparisons with currently used models are difficult. However, based on the results shown in this work, a move from the current $10\,\mathrm{km}$ resolution sea ice models (Ólason et al., 2021; Hutter et al., 2022) to practical kilometer-scale models seems tangible.

*Code and data availability.* A snapshot with all the code needed to reproduce the experiments in this manuscript is available on Zenodo (Jendersie et al., 2024). The input data for the realistic case is provided seperatly (Jendersie and Spain, 2025). The project neXtSIM-DG is under active development and hosted on GitHub (https://github.com/nextsimhub/nextsimdg). The full model code provided with this paper is based on v0.3.1 of neXtSIM-DG, with minor bug fixes and added GPU support which is not part of the main release. The provided standalone dynamical core, used for the comparison of the different GPU implementations of the stress, is also available as a self contained repository at https://github.com/nextsimhub/nextsimdg. The relevant versions are tagged as v0.3.1b.

*Author contributions.* Robert Jendersie developed the GPU codes, conducted the experiments and wrote the bulk of the text. Christian Lessig gave substantial input on the research direction and analysis methods and worked on the final text. Thomas Richter provided the original CPU code as well as the model description and helped to improve the final text.

*Competing interests.* The authors declare that they have no conflict of interest.

*Acknowledgements.* This project is supported by Schmidt Sciences. The authors gratefully acknowledge the Gauss Centre for Supercomputing e.V. (www.gauss-centre.eu) for funding this project by providing computing time on the GCS Supercomputer JUWELS (Jülich Supercomputing Centre, 2021) at Jülich Supercomputing Centre (JSC). We acknowledge the EuroHPC Joint Undertaking for awarding this project access to the EuroHPC supercomputer LUMI, hosted by CSC (Finland) and the LUMI consortium through a EuroHPC Regular Access call.

## Appendix A: Introduction to GPUs

Graphics processing units are coprocessors originally designed to accelerate computer graphics. To generate images in quick succession, a massive number of operations, e.g. computing the color of each pixel on the screen, has to be performed. The

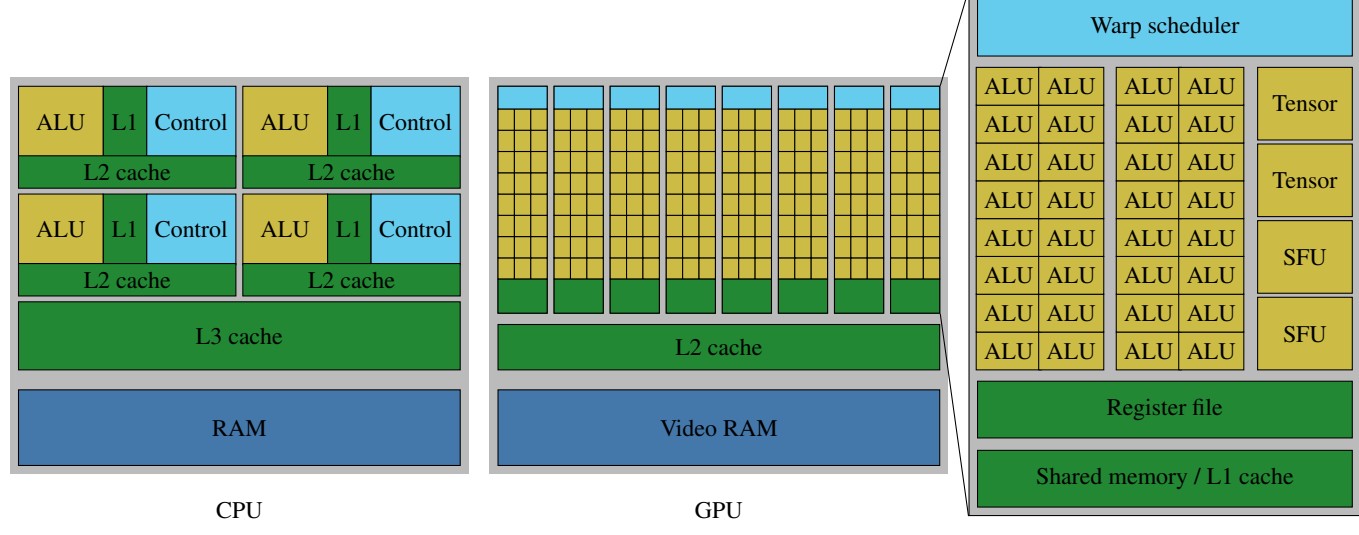

**Figure A1.** A schematic overview of an example CPU versus a GPU.

computations are typically identical and it is thus a highly parallel problem. Initially, the specific algorithms used for this purpose were implemented directly in the hardware and the resulting *fixed function pipeline* offered only very limited possibilities for customization. However, to enable higher fidelity graphics, most parts of the pipeline where replaced with programmable stages over time. Attempts to use such programmable GPUs for scientific computing soon followed (Bolz et al., 2003; Krüger and Westermann, 2005). To further facilitate the application of GPUs beyond computer graphics, new programming interfaces where developed that represent GPUs as general purpose *stream processors* (Buck et al., 2004). In this paradigm, a program takes a long sequence (a stream) of data and applies *kernel functions* to each element in the sequence. Among the general purpose programming interfaces, CUDA (NVIDIA, 2023a) has been the most influential and the fundamental programming model defined by CUDA has stayed the same since its introduction in 2006.

In the following, we give an overview to GPUs as general purpose parallel processors, aimed at readers somewhat familiar with the architecture of CPUs. In the interest of brevity, we only use the nomenclature of CUDA, but the concepts and components described here also exist on GPUs of other vendors, sometimes under different names. GPUs are designed for high throughput and can, given an embarrassingly parallel problem, deliver significant gains in performance and energy efficiency over CPUs. Naturally, this requires a trade-off, since chip resources need to be distributed differently. As Fig. A1 shows, for CPUs, much of the die space is allocated to the multi-level cache hierarchy and control circuits, e.g. speculative execution, with the goal to minimize latency as much as possible. In contrast, GPUs prioritize compute throughput with many more arithmetical logical units (ALUs) and a much wider execution pipeline. If sufficient work is available to saturate the compute pipeline, latency can, in principle, be hidden because computations in different stages of execution can keep every part of the hardware busy. For a program to take advantage of just a single GPU, it needs thousands of similar computations that can

be run in parallel, as opposed to just a few tasks needed to fully utilize a multi-core CPU. In practical terms, a good balance between operations is required to achieve high utilization, i.e. the compute units can only be kept busy if the data can be loaded and stored with the same speed as it is processed. The dedicated main memory on GPUs (VRAM) is therefore also optimized toward higher bandwidth whereas regular RAM for CPUs prioritizes lower latency.

To access data on the GPU it needs to be first moved to *global memory*, a logical space residing in the VRAM. Since transfers between the host memory (system RAM), and device memory (VRAM) are slow, the work offloaded to GPUs needs to be sufficiently expensive. Therefore, when a complex computation is performed, it can be worthwhile to perform an individual operation on the GPU, even if it is not efficient, when this minimizes the data transferred between GPUs and CPUs.

    The main building block of a GPU is the **streaming multiprocessor (SM)**. Equipped with its own cache, registers, ALUs 625 and control logic, a streaming multiprocessor can operate largely autonomously and is similar to a core on the CPU. The number of streaming multiprocessors varies widely between chips and performance considerations for a program with respect to this number are similar to varying the number of CPU cores, i.e. weak and strong scaling. High-end GPUs have around 128 streaming multiprocessors.

    Conceptually, each streaming multiprocessor is a single instruction, multiple data (SIMD) processor that achieves computa- 630 tional efficiency by data parallel processing, i. e. through a number of threads that perform the same computation on different data. Such a **warp** of threads is comparable to a CPU thread where every operation is performed as SIMD. However, warps are much wider than the vector units commonly found on CPUs. Figure A1 alludes to this with the ALUs (or *CUDA cores*) being grouped in sets of 16, though in practice, a common warp size is 32. As a consequence of the SIMD paradigm, branches ("if" statements) can be hugely detrimental to the performance on the GPU if they cause frequent *warp divergence*, i.e. the selection 635 of different code paths within a warp. In that case, the whole warp will execute each taken branch and undesired results are just masked out at the end. A similar problem exists for CPU code, where branches can prevent effective vectorization.

    Each GPU thread has its own set of registers and private *local memory* if additional space is needed. Since local memory is just a special address space in VRAM, it has very high latency, and it puts additional strain on the same memory bus as global memory accesses. While automated caches alleviate the issue of slow device memory to some extent, much better performance 640 can often be achieved by utilizing **shared memory**. This memory space is part of the L1 cache, allowing for fast random access, but is manually programmed so that one can ensure that the right data is held in the cache. The same shared memory space is shared between multiple threads.

    Multiple warps together form a **thread block** and all threads in the same block are resident on a single streaming multi- processor at the same time. Instead of having a fixed number of registers available to each thread, on GPUs, the registers are 645 allocated as needed for each kernel from the **register file**. Keeping the state of multiple warps in registers is necessary to make context switches between them cheap, which is key to achieve a high throughput. A high per-thread register requirement can therefore be detrimental to performance, because it decreases the possible thread block size. A lower *occupancy* means that fewer warps are available to the scheduler, which in turn increases the likelihood of wasted cycles. An important feature of thread blocks is their ability to effectively coordinate work. Synchronization is possible via a lightweight barrier and all threads 650 in a block have access to the same shared memory, thereby making it possible to share intermediate results.

All thread blocks are organized into a **grid** and independently executed on the available streaming multiprocessors. Synchronization between blocks usually happens only once the whole grid is finished, although another intermediate level called *cooperative groups* is available on the latest hardware.

In addition to the general purpose ALUs, streaming multiprocessors are equipped with a number of fixed function units to accelerate specific computations. These include **tensor cores** which compute matrix-matrix multiplications and **special function units (SFUs)** that compute approximations for certain transcendental functions. Programming with tensor cores requires special consideration because tensor cores are controlled at the warp level, whereas the program is written in terms of threads.

## Appendix B: Code

```
template <DG> using DGVec = Eigen::Matrix<T, Eigen::Dynamic, DG>;
template <int CG, int DGstress, int DGadvection>
void StressUpdateHighOrder(const VPParameters& vpparameters,
const ParametricMomentumMap<CG>& pmap, const ParametricMesh& smesh,
DGVec<DGstress>& S11, DGVec<DGstress>& S12, DGVec<DGstress>& S22,
const DGVec<DGstress>& E11, const DGVec<DGstress>& E12, const DGVec<DGstress>& E22,
const DGVec<DGadvection>& H, const DGVec<DGadvection>& A, double alpha, double beta)
{
constexpr int NGP = ((DGstress == 8) || (DGstress == 6)) ? 3 : (DGstress == 3 ? 2 : -1);
using EdgeVec = Eigen::Matrix<T, 1, NGP * NGP>;
#pragma omp parallel for
for (size_t i = 0; i < smesh.nelements; ++i) {
auto hGauss = (H.row(i) * PSI<DGadvection, NGP>).array().max(0.0).matrix();
auto aGauss = (A.row(i) * PSI<DGadvection, NGP>).array().max(0.0).min(1.0).matrix();
EdgeVec P = (_vpparameters.Pstar * hGauss.array()
* (-20.0 * (1.0 - aGauss.array())).exp()).matrix();
const EdgeVec e11Gauss = E11.row(i) * PSI<DGstress, NGP>;
const EdgeVec e12Gauss = E12.row(i) * PSI<DGstress, NGP>;
const EdgeVec e22Gauss = E22.row(i) * PSI<DGstress, NGP>;
const auto DELTA = (vpparameters.DeltaMin * vpparameters.DeltaMin
+ 1.25 * (e11Gauss.array().square() + e22Gauss.array().square())
+ 1.50 * e11Gauss.array() * e22Gauss.array() + e12Gauss.array().square())
.sqrt().matrix();
const T alphaInv = 1.0 / alpha;
const T fac = 1.0 - alphaInv;
const EdgeVec PDelta = P.array() / DELTA.array();
S11.row(i) = fac * S11.row(i) + (pmap.iMJwPSI[i]
* (alphaInv * (PDelta.array()
* ((5.0 / 8.0) * e11Gauss.array() + (3.0 / 8.0) * e22Gauss.array())
33                 - 0.5 * P.array()).matrix().transpose())).transpose();
S12.row(i) = fac * S12.row(i) + (pmap.iMJwPSI[i]
* (alphaInv * (PDelta.array() * (1.0 / 4.0) * e12Gauss.array())
.matrix().transpose())).transpose();
S22.row(i) = fac * S22.row(i) + (pmap.iMJwPSI[i]
* (alphaInv * (PDelta.array()
* ((5.0 / 8.0) * e22Gauss.array() + (3.0 / 8.0) * e11Gauss.array())
40                 - 0.5 * P.array()).matrix().transpose())).transpose();
}
}
```

**Listing 2.** Implementation of the stress update with Eigen. The method is generic in the degrees of freedom of the different cG and dG elements. The `matrix()` and `array()` methods change the type of an expression to differentiate between matrix and component-wise operations and are no-ops during runtime.

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
