# Peer review of "A GPU-parallelization of the neXtSIM-DG dynamical core (v0.3.1)"

_EGUsphere, 2024_

## Referee Comment (RC1)

Review "A GPU-parallelization of the neXtSIM-DG dynamical core

In general a nice and well written manuscript. It runs through a number of general purpose GPU methods for optimization of the stress part of mEVP and inclusion of this into neXtSIM-DG. The manuscript  also look into the how well these frameworks can be transferred to CPU's which I think is important as well as the performance. The advantage Is that the frameworks are relatively easy to plugin. The disadvantage is that it is a black box to some extend.

The one thing that I miss the most is a real life example and not just the theoretical example. One of the challenges in traditional sea ice models is the inclusion of land and ocean points, where active sea ice points may be a minimal fraction of the total number of points. If I remember correct neXtSIM-DG has a fixed grid which could potentially leave a significant number of points icefree/inactive. I think that this at least deserves some comments and ideally a testcase. I do realize that running a new test case may be a bit out of scope.

The mEVP and the neXtSIM/Elasto-Brittle seems to be mixed. It may be true that the dynamics of neXtSIM are similar in the numerical sense but it does represent two different variation of sea ice dynamics. If neXtSIM has reverted to use mEVP (I don't think is the case) then the refactorization within this paper is still of interest. I just don't understand why this is investigated. For instance in table 1 timings of mEVP?  It would be good if the authors put in a comment why the mEVP method has been refactored instead of the Elasto Brittle if this is the case. Alternatively please call the rheology something else than mEVP if this is not the rheology that is being refactored and describe the relevant rheology.

There are places like  introduction of the mEVP solver and the discetization where the authors introduce it but cut it off very quickly with a reference.  A bit more elaboration would help.

In addition the manuscript opens several topics for discussion and further development throughout the paper. This leaves the reader with the thought that optimization with the methods described here requires some soft of re-evaluation every time a new computer is used. Is this correct or not?

Abstract:
I would leave out the comment about energy in line 3. It is definitely of interest and important but it is not really the scope of this manuscript. It would be preferable with a conclusion in the abstract if this is deemed important enough to be mentioned in the abstract.

Introduction

Line 24: I am not sure that the order of the solver is the biggest inaccuracy of the solution in a climate model. I would add "numercial" in front of accuracy.

Line 34: I would not include stand-alone systems in this context as these are mostly used demonstrators and test as it is the case in the manuscript. The model would almost always be run as part of a coupled system, especially for longer simulations.

Line 41 - 45: Energy is mentioned in the abstract. If that is kept it should also be mentioned here.

Line 74 Section Sect. 2: I would remove Sect.

 Line 96 and the first line in 2.1 is a repetition. I would remove one of them.

Line 99: Mapping of spherical coordinates (lat,lon?) to computational domain. A bit more description of this would be good. Is it also quadrilateral? Geophysical models refer to "lat/lon" but none of them use it for the computational grid. Which shape (if any specific ) is the computational grid on?

Comparison of CPU vs GPU: Are all comparisons based on a 10cpu run?
Is the cpu runs executed on System 1?
One can always compare the compute time but it is difficult to compare an "openMP" code run on CPU directly with a GPU simulation.

Table 2 It would be beneficial to add the standard deviation of these timings in order to show if the difference in runtime is significant compared to the difference of the same run.

Section 3
This section explores a number of different optimization efforts. These are all carried out for 1 setup. The only viable solution is CUDA and KOKKOS. The rest fail due to crashes, limitation when using EIGEN.  It is a concern that the use of Eigen code crashes when used within OpenMP/OpenACC. An explanation is found with the SYCL code. This leads to a question whether the choice of EIGEN should be reconsidered?

CUDA+KOKKOS: Has the authors tested if the results are different for different domains and resolutions? The memory available is described to be less than half of what is available. Does this imply that if the resolution were doubled then the memory access would be important? Should a new refactorization be carried out then?
Are the results bit for bit (probably not) when compared to openMP/single processor solutions? Do they change significantly? This has partly been checked later in the study regarding precision.

Section 4
I think that the description of the numerical experiment should be before the description of the different optimization methods as timings of the experiments are used. If I read correct it is the largest experiment that was used for section 3. Is that correct? It could be made more clear.

Figure 2: It would be helpful if the cpu runs had the same symbol or line style (e.g. square symbol or dashed line) and the GPU's was marked in a different way.

Line 329: Ice models are often a part of an Earth System model

Line 357: The fact that this is an argument make a direct comparison impossible.

Figure 5. I am not entirely sure what the Noise amplitude is on the x axis. Please clarify.

Line 389: It is true that if I could get observation of concentration and thickness within 2% then I would be very happy. This uncertainty varies a lot depending on location and time of year. The example of a one day run satisfy one end of the time scales, however it does not say if this is good enough for e.g. longer climate series, where years are simulated. Conclusions on this should be formulated weak.

Line 397. What is the expected benchmark? This paragraph needs a few references or descriptions of why the speedup is anticipated to increase differently.

Line 425: If such a slowdown is imposed by higher order methods. Are they then worth the effort?

Line 491: This should also be considered on CPU's

Line 506. These runs are already at ~5km and lower. It would be nice if the authors elaborated a bit on the different test domains, how the compare to reality and the influence on the results.

Table 4: Please specify ns, ng and na. Can this be moved closer to line ~440 where it is used.

Figure 8+ implementation within : Why is Cuda numbers presented if the implementation in section 5 is Kokkos

---

## Author Response (AR1)

Dear editor,

We have taken all the remarks into account and we believe that they helped us to improve the quality of our manuscript. Below we reply to each reviewer comment and make note of changes done for each point raised. We hope that our revisions adequately address the reviewer's suggestions.

Best regards,

Robert Jendersie, on behalf of the authors

**Review 1**

*In general a nice and well written manuscript. It runs through a number of general purpose GPU methods for optimization of the stress part of mEVP and inclusion of this into neXtSIM-DG. The manuscript also look into the how well these frameworks can be transferred to CPU's which I think is important as well as the performance. The advantage Is that the frameworks are relatively easy to plugin. The disadvantage is that it is a black box to some extend.*

*The one thing that I miss the most is a real life example and not just the theoretical example. One of the challenges in traditional sea ice models is the inclusion of land and ocean points, where active sea ice points may be a minimal fraction of the total number of points. If I remember correct neXtSIM-DG has a fixed grid which could potentially leave a significant number of points icefree/inactive. I think that this at least deserves some comments and ideally a testcase. I do realize that running a new test case may be a bit out of scope.*

**Reply:** Due to the choice of the grid, the impact of land areas on the performance is indeed limited in neXtSIM-DG. In principle, the parametric meshes can be adjusted in order to maximize the share of active mesh cells. Where the geometry makes this difficult, land elements do incur a similar cost to active cells. While most computations are not performed for land cells, it is hard to take full advantage of this reduced workload in parallel, because the runtime is determined by the slowest thread. This is an import trade-off of our numerical model that deserves some consideration.
**Changes:** We added a more realistic test case of the arctic in Sect. 5 to discuss the effects of the land mask and the coordinate system and how GPU and CPU versions are impacted differently.

*The mEVP and the neXtSIM/Elasto-Brittle seems to be mixed. It may be true that the dynamics of neXtSIM are similar in the numerical sense but it does represent two different variation of sea ice dynamics. If neXtSIM has reverted to use mEVP (I don't think is the case) then the refactorization within this paper is still of interest. I just don't understand why this is investigated. For instance in table 1 timings of mEVP? It would be good if the authors put in a comment why the mEVP method has been refactored instead of the Elasto Brittle if this is the case. Alternatively please call the rheology something else than mEVP if this is not the rheology that is being refactored and describe the relevant rheology.*

**Reply:** NeXtSIM-DG is a flexible framework to model sea ice. It is versatile in design and it supports several rheologies with a focus on both viscous-plastic and brittle rheologies. The mEVP solver for VP is, in its algorithmic setup, nearly identical to the BBM rheology. They share the same momentum equation and have the same structure with costly sub-stepping involving the computation of the strain rate tensor and the divergence of the stresses. For the computational aspects that we study, the rheology largely does not matter. The shape of our data and operations is determined by the chosen discretization. We decided to base the presentation on the mEVP solver as it is more established and widely used in the community such that the description of a complex rheology does not distract from the actual content, the GPU implementation.
**Changes:** We added a paragraph to the introduction about neXtSIM-DG to clear up this confusion.

*There are places like introduction of the mEVP solver and the discetization where the authors introduce it but cut it off very quickly with a reference. A bit more elaboration would help.*

**Reply:** We will add more details in Sect. 2. to make the paper more self-contained.

**Changes:** We added more details in Sect. 2 and Sect. 2.1.

*In addition the manuscript opens several topics for discussion and further development throughout the paper. This leaves the reader with the thought that optimization with the methods described here requires some soft of re-evaluation every time a new computer is used. Is this correct or not?*

**Reply:** This is true to some extent, since GPU architectures are constantly evolving with new features being added in each generation. Nonetheless, the "fast path" of computations on GPUs has been largely unchanged for many years. Furthermore, with continued developments in software (e.g. heterogeneous compute frameworks) and hardware (e.g. well-tuned automated caches) it is becoming less of an issue. For our implementation we found that we do not need specialized features to achieve good performance, which makes the code portable. This assertion is supported by further tests done during the development on other GPUs (V100, H100, RTX 3090) in addition to those reported on in this manuscript.
**Changes:** The results presented for the new test case are from different hardware, providing an additional data point for the portability.

*Abstract*
*I would leave out the comment about energy in line 3. It is definitely of interest and important but it is not really the scope of this manuscript. It would be preferable with a conclusion in the abstract if this is deemed important enough to be mentioned in the abstract.*

**Reply:** Agreed, since we do not emphasize the energy aspect elsewhere we will leave it out of the abstract.
**Changes:** We removed the energy aspect.

*Introduction*
*Line 24: I am not sure that the order of the solver is the biggest inaccuracy of the solution in a climate model. I would add "numercial" in front of accuracy.*

**Reply:** Agreed, "numerical accuracy" is more precise here.
**Changes:** The wording was changed.

*Line 34: I would not include stand-alone systems in this context as these are mostly used demonstrators and test as it is the case in the manuscript. The model would almost always be run as part of a coupled system, especially for longer simulations.*

**Reply:** We agree with the reviewer that a sea ice model like ours is typically used in a larger coupled climate model. Nonetheless, to study the model itself it is best run in a standalone mode, as is common practice in our opinion. Furthermore, the emphasis on standalone models was suggested to us by a colleague, so there is at least some interest in the sea-ice community. For NeXtSIM-DG in particular, the impact of the modified dynamics and the choice of the discretization warrant further investigation of just the sea-ice model. We would therefore keep this as part of our motivation.
**Changes:** No changes.

*Line 41 - 45: Energy is mentioned in the abstract. If that is kept it should also be mentioned here.*

**Reply:** We will leave the energy aspect out.
**Changes:** No changes.

*Line 74 Section Sect. 2: I would remove Sect.*

**Reply:** Good catch. In line with the journal guidelines we will remove the non-abbreviated "Section".
**Changes:** Fixed.

*Line 96 and the first line in 2.1 is a repetition. I would remove one of them.*

**Reply:** Well keep this in mind when updating Sect. 2.
**Changes:** The reference was moved to the beginning of Sect. 2 and the repeated sentence removed.

*Line 99: Mapping of spherical coordinates (lat,lon?) to computational domain. A bit more description of this would be good. Is it also quadrilateral? Geophysical models refer to "lat/lon" but none of them use it for the computational grid. Which shape (if any specific ) is the computational grid on?*

**Reply:** Yes, the mesh consists of topologically regular quadrilaterals, forming a rectangular domain. The parametric map takes the reference cell and maps it to each distorted cell on the sphere. This is discussed in greater detail in Richter et al. (2023a), but we will add additional details in Sect. 2 to make the manuscript more self-contained.
**Changes:** Additional details on the mesh and coordinates where added in Sect. 2.1.

*Comparison of CPU vs GPU: Are all comparisons based on a 10cpu run? Is the cpu runs executed on System 1?*

**Reply:** Only the timings in Table 1 are from the 10 core cpu. All other CPU results are from System 1.
**Changes:** We added explicit mentions of the system used in captions where it was missing.

*One can always compare the compute time but it is difficult to compare an "openMP" code run on CPU directly with a GPU simulation.*

**Reply:** We agree that a direct comparison can be difficult, and that CPU/GPU balance can be complicated in a large coupled simulation. Nonetheless, simulation-years-per-computer-day is a common performance measure, which is often considered as the user-oriented output metric. This is what we follow in our work.
**Changes:** No changes.

*Table 2 It would be beneficial to add the standard deviation of these timings in order to show if the difference in runtime is significant compared to the difference of the same run.*

**Reply:** Sure, we can add the standard deviation to the table. The deviations between repeated runs are fairly small for GPU measurements, ranging from 0.001 to 0.004 (at most 1%).
**Changes:** The standard deviation was added to the table.

*Section 3*
*This section explores a number of different optimization efforts. These are all carried out for 1 setup. The only viable solution is CUDA and KOKKOS. The rest fail due to crashes, limitation when using EIGEN. It is a concern that the use of Eigen code crashes when used within OpenMP/OpenACC. An explanation is found with the SYCL code. This leads to a question whether the choice of EIGEN should be reconsidered?*

**Reply:** Eigen is a popular choice for linear algebra in C++ for good reasons. Its powerful abstractions enable us to write concise and efficient code that is generic in the discretization order and the datatype. Any option providing comparable features would face similar problems when used for GPU programming. Since we found frameworks that do work well with Eigen, we consider Eigen viable for future development. Furthermore, due to its popularity, Eigen's GPU will likely improve further in the feature. For example, improved SYCL compatibility is being worked on `https://gitlab.com/libeigen/eigen/-/issues/2763`).
**Changes:** No changes.

*CUDA+KOKKOS: Has the authors tested if the results are different for different domains and resolutions? The memory available is described to be less than half of what is available. Does this imply that if the resolution were doubled then the memory access would be important? Should a new refactorization be carried out then?*

**Reply:** So far we have not done extensive testing on different domains, but the required memory depends only on the number of elements which is covered by our resolution scaling test. While the largest domain tested with $1.7 \times 10^7$ does fit in the 40GB VRAM available on the A100, the device memory can certainly be a limiting factor for higher orders and when even more work is done on the device. Additional effort may be needed to optimize memory usage in the future. Alternatively, as long as we can fit $10^6$ elements in VRAM, a single GPU should be well utilized and further scaling can also be made possible by supporting multiple GPUs. In the manuscript, we currently only make note of the constant memory usage (with less than half used), which is independent of the number of elements and therefore not an issue. We will present data on the device memory usage and discuss the impact of different domains and resolutions in the revised manuscript.
**Changes:** A discussion on the memory usage was added in Sect. 5.

*Are the results bit for bit (probably not) when compared to openMP/single processor solutions? Do they change significantly? This has partly been checked later in the study regarding precision.*

**Reply:** While the accuracy is essentially the same, there are deviations of the order of machine precision for computations run on the GPU. Even without fast-math like flags enabled, the GPU compiler can make different decisions on fused-multiply-add operations, which are more accurate than the separate operations. Furthermore, kernels that rely on atomics for synchronization are nondeterministic, i.e. results will differ slightly for each run even on a single machine. Combined with the high sensitivity of the model, investigated in Sect 4.2, this means that while results are qualitatively the same, there could be visible differences over longer time periods.
**Changes:** The note on the accuracy at the beginning of Sect. 4 was rephrased to mention this aspect.

*Section 4*
*I think that the description of the numerical experiment should be before the description of the different optimization methods as timings of the experiments are used. If I read correct it is the largest experiment that was used for section 3. Is that correct? It could be made more clear.*

**Reply:** Optimizations where evaluated on a domain with $2.6 \times 10^5$ elements with the setup described in Sect. 4, which is the same as the 3rd smallest experiment. We will add this information to the table. Other details of the setup are not relevant for the optimizations discussed in Sect. 3 and therefore, in our opinion, better left for Sect. 4. The stress kernel operates independently on each element and there is no special treatment for land or sea-ice cells, so the distribution of

land has no effect. Furthermore, the cell-indexing and the shape of the thread-blocks is 1D in all experiments, meaning that the memory access patterns are the same regardless of the concrete shape of the rectangular mesh. **Changes:** We kept the order in our manuscript but briefly mention that the benchmark is used in Sect. 3.

*Figure 2: It would be helpful if the cpu runs had the same symbol or line style (e.g. square symbol or dashed line) and the GPU's was marked in a different way.*

**Reply:** This is a good idea. We will change the line-style for CPU results.
**Changes:** The plots where updated. Additionally, we added (CPU) or (GPU) to the legend entries where appropriate to further clarify the distinction.

*Line 329: Ice models are often a part of an Earth System model*

**Reply:** This comment appears to be incomplete.
**Changes:** No changes.

*Line 357: The fact that this is an argument make a direct comparison impossible.*

**Reply:** The difference in the measurement methods should have a limited impact on the results. While the kernel launch overhead may not be fully accounted for in the AdaptiveCPP measurement, this cost is independent of the problem size and therefore only relevant for a small simulation. Since the GPU implementation is mainly useful for large simulations, and if anything, the measurement bias is in favor of AdaptiveCPP, the larger conclusion is not affected. For the revised manuscript, we will clarify this caveat.
**Changes:** This detail was added in Sect 4.1 (now Line 380).

*Figure 5. I am not entirely sure what the Noise amplitude is on the x axis. Please clarify.*

**Reply:** We run the benchmark repeatedly with modified initial conditions. In each run, the initial ice height and concentration fields are perturbed by adding a random value to each degree of freedom. These values are sampled uniformly from the interval $[-\epsilon, \epsilon]$. On the x-axis, we vary the maximum size of the perturbation. For each chosen $\epsilon$ and float type, we run 24 simulations with noise of that magnitude. We will update the description of Figure 5 to make the experimental setup more clear.
**Changes:** The caption of Figure 5 was rewritten.

*Line 389: It is true that if I could get observation of concentration and thickness within 2% then I would be very happy. This uncertainty varies a lot depending on location and time of year. The example of a one day run satisfy one end of the time scales, however it does not say if this is good enough for e.g. longer climate series, where years are simulated. Conclusions on this should be formulated weak.*

**Reply:** The topic of mixed precision and uncertainties certainly warrants further study, but this is outside the scope of our current work. We consider the semi-realistic perturbations useful as an indicator for this real-world issue, though we will make sure to mention the limited timescale of our results.
**Changes:** Both in Sect. 4.2 and in the conclusion (Sect. 6) now mention this.

*Line 397. What is the expected benchmark? This paragraph needs a few references or descriptions of why the speedup is anticipated to increase differently.*

**Reply:** The key performance metrics, compute and memory throughput, are effectively doubled for F32, so a factor two speedup is expected. However, other factors, such as occupancy and the

usage of special function units, also change with the switch to F32 and their impact is harder to predict. We do not anticipate major differences for Kokkos and AdaptiveCPP compared to CUDA, as they are structurally similar. For AdaptiveCPP, with its different compiler, the other factors could, however, play a larger role. We will add these considerations to the referenced paragraph.

**Changes:** The paragraph (now starting at Line 412) was extended.

*Line 425: If such a slowdown is imposed by higher order methods. Are they then worth the effort?*

**Reply:** At present, we do aim to primarily use the higher order method. However, to properly answer this question, a more thorough investigation of the trade-off between speed and accuracy is needed, which is left for future work. Our analysis only shows that this trade-off does not fundamentally change with the switch to GPUs. Nonetheless, recent benchmark computations comparing different discretizations demonstrate the superiority of higher order methods (Mehlmann et al., 2021; Richter et al., 2023b). We will add a remark and citations.

**Changes:** A remark was added in Sect 4.3 (Line 458).

*Line 491: This should also be considered on CPU's*

**Reply:** In principle yes, but since we only have F32 performance measurements for GPU, we can't draw the same conclusions for CPUs. Especially if the code is not fully vectorized, we expect the benefit of F32 to be smaller on CPU. We leave this for future work.

**Changes:** No changes.

*Line 506. These runs are already at 5km and lower. It would be nice if the authors elaborated a bit on the different test domains, how the compare to reality and the influence on the results.*

**Reply:** The grid spacing is not important in our scaling test. It is just a convenient way to increase the number of elements, since from a compute perspective, the results are the same as a larger domain as long as the simulation remains stable. To properly discuss other aspects that change with the domain, we will add a more realistic test case.

**Changes:** We added the arctic test case in Sect 5.

*Table 4: Please specify ns, ng and na. Can this be moved closer to line 440 where it is used.*

**Reply:** We will add a variable description and improve the placement of the table.

**Changes:** The placement was improved.

*Figure 8+ implementation within : Why is Cuda numbers presented if the implementation in section 5 is Kokkos*

**Reply:** During the initial evaluation, CUDA was the reference implementation and simply had the most features implemented, including the order switching. Given that the CUDA and Kokkos kernels are essentially the same under the hood, the conclusions drawn from this experiment should apply to our proper Kokkos implementation as well.

**Changes:** No changes.

**Review 2**

*In the present manuscript 'A GPU-parallelization of the neXtSIM-DG dynamical core (v0.3.1)' the authors test and evaluate different GPU programming frameworks based on their sea ice model*

*dynamical core neXtSIM-DG.*

*Many modeling groups in the weather and climate community and beyond are facing similar problems as the neXtSIM-DG developers. Developing portable code that achieves good performance on various hardware architectures without limiting the productivity of the (scientific) developers too much is a major challenge. Therefore, the thorough analysis of the different available GPU programming frameworks presented here is of great value to the community. The study is well written and I would recommend publication in GMD after a few issues have been addressed as listed below.*

1. *In line 370 in section 4.1 it is stated 'These results indicate that the AMD ecosystem is still less mature'. However, to validate this statement and to have a complete picture also for AMD GPUs it would have been nice to also have a HIP implementation as a baseline to compare the other implementations against similar to the CUDA implementation for NVIDIA GPUs.*

   **Reply:** A raw HIP implementation would certainly further substantiate this point. However, it is reasonable to expect that in our case a raw HIP implementation would perform similar to the Kokkos implementation. We ended up using only basic features of Kokkos in the main comparison, which are provided through light wrappers around the vendor specific APIs. Considering how our CUDA and Kokkos implementations perform the same asymptotically and how closely the basic HIP API resembles that of CUDA, a raw HIP implementation would likely display similar characteristics.

   **Changes:** No changes.

2. *Table 1: What hardware was used for these measurements and how many OpenMP threads were used?*

   **Reply:** These measurements where taken on an Intel i9-10900X (10 cores @ 3.7GHz). For OpenMP we determined that simultaneous multithreading is beneficial and used 20 threads. We will add these details to the table.

   **Changes:** The table caption was updated.

3. *Figures 2 and 10 and lines 354 and 470: Again, how many OpenMP threads were used for the OpenMP reference simulation? And what backend was used for Kokkos on CPUs? OpenMP as well? And if yes, with the same number of threads as the reference OpenMP simulation?*

   **Reply:** We got the best performance through full utilization with simultaneous multithreading, i.e. 96 threads, with `OMP_PROC_BIND=spread` and `OMP_PLACES=threads`. For Kokkos (CPU) we use the OpenMP backend with the same settings. We will add this information to the manuscript.

   **Changes:** Additional details on the OpenMP configuration where added where appropriate in Sect. 4 and Sect. 5.

4. *Line 206: LLVM/Clang provides a set of debugging flags (e.g. https://openmp.llvm.org/design/Runtimes.h info) which can provide precise information about each block of memory and potential problems. Also, for the types that are not trivially copyable, OpenMP 5.0 offers the option of using declare mapper to define this. Wouldn't that have been an option here?*

   **Reply:** These are good suggestions. With the diagnostics provided by Clang, a working OpenMP offload implementation would likely be doable. Unfortunately, declare mapper would not help much with memory transfers, since it is only suited for C-style code. Almost all data in our code is held in Eigen::Matrix objects. This type is generic (template

parameters include size, data type and storage order) and has a variable memory layout. Depending on whether the size is fully known at compile time or not, the data can be part of the struct or a pointer to the heap. Both data pointers and dynamic size are private members and need to be accessed through method calls, which, as far as I can tell, are not allowed in declare mapper directives. So one would still end up manually converting the buffers to a simpler structure for transfers.

**Changes:** No changes.

*Technical corrections:*

- *Table 2: 'AdaptiveCPP' is used here to indicate the SYCL implementation but the name is too generic. AdaptiveCPP is also the name of the compiler and it can also compile native OpenMP or other parallel APIs. I would suggest replacing 'AdaptiveCPP' with 'SYCL-AdaptiveCPP'.*
  **Reply:** Makes sense. We will rename 'AdaptiveCPP' to 'SYCL-AdaptiveCPP' in the other sections as well to be consistent.

- *Figure 2: Why is in the legend of the right panel TorchInductor marked with an '*'?*
  **Reply:** These measurements had a small inaccuracy and that would have warranted a comment. In the revision we will update the data. The changes are minor.

- *Line 16: impact on long-term processes*

- *Line 42: is -¿ it*

- *Line 88: often often -¿ Remove one*

**Changes:** The listed corrections where integrated. The data for TorchInductor in Fig. 2 was updated.

**References**

Mehlmann, C., Danilov, S., Losch, M., Lemieux, J. F., Hutter, N., Richter, T., Blain, P., Hunke, E. C., and Korn, P.: Simulating Linear Kinematic Features in Viscous-Plastic Sea Ice Models on Quadrilateral and Triangular Grids With Different Variable Staggering, Journal of Advances in Modeling Earth Systems, 13, https://doi.org/10.1029/2021ms002523, 2021.

Richter, T., Dansereau, V., Lessig, C., and Minakowski, P.: A dynamical core based on a discontinuous Galerkin method for higher-order finite-element sea ice modeling, Geoscientific Model Development, 16, 3907–3926, https://doi.org/10.5194/gmd-16-3907-2023, 2023a.

Richter, T., Dansereau, V., Lessig, C., and Minakowski, P.: A snippet from neXtSIM_DG next generation sea-ice model with DG, https://doi.org/10.5281/ZENODO.7688636, 2023b.

---

## Author Response (AR2)

Dear editor,

Thank you for accepting our manuscript.

We updated the code and data availability statement to clarify how the different code versions relate. The full model code presented in the paper is based on the release 0.3.1 (GitHub) with added GPU support which is currently not part of the development releases. Furthermore, minor bug fixes where back ported to properly run the new case, thus version 0.3.1b. The standalone dynamical core with the different GPU implementations is also part of the Zenodo release. To highlight the corresponding versions in the Gitlab repository, we added tags to the relevant commits.

The final comments of the reviewer have been integrated into the final version of the manuscript:

*Reviewer: Line 12: Line starting with additionally. I would either add a short conclusion on the study of the higher order numerical schemes as it is done with the numerical precision. Otherwise, I would leave this out.*

We left out the first part on higher-order discretizations.

*Reviewer: Line 15 cyrospherre = cryosphere*

We fixed the typo.

*Reviewer: Line 19: This refers to the atmospheric models? Maybe mention that.*

We clarified in the text that this refers to all Earth system components.

*Reviewer: Line 77 – 84 This looks like something from a summary, discussion or abstract. Consider to move it*

We decided to leave a summary of the results in the introduction to provide the reader with a teaser of the conclusions he can expect.

*Reviewer: Line 100: This starts as a general "often" and becomes specific with namely. I would rephrase to say that "here the mEVP solver is used" instead of namely (or something like that). There are other flavors of the solution than the mEVP.*

We adapted this as suggested by the reviewer.

*Reviewer: Line 475: Without having read the two references, then the lower computational efficiency will lead to reduction in e.g. number of iteration, resolution etc. Then it is not sure that the total solution is better despite that the higher order solver seen isolated improves the solution. This is also discussed in the conclusion (line 584 – 585).*

We added a comment to clarify the point.

*Reviewer: Figure 10+11? Is it correctly understood that section 3.2 close the use of oopenMP/openACC for GPU but not CPU's?*

Yes, as discussed in Sec. 3.2 in our manuscript, OpenMP/OpenACC were not usable for us on the GPU.

*Line 512 (continuation of point before): This is mentioned below. It seems to be the GPU version of Kokkos that is faster not the CPU version. I would soften this and state that the gpu*

We clarified that the GPU version of Kokkos is meant.

Best regards,

Robert Jendersie, on behalf of the authors